



# Documentary data and the study of the past droughts: an overview of the state of the art worldwide

Rudolf Brázdil[1,2], Andrea Kiss[3,4], Jürg Luterbacher[5,6], David J. Nash[7,8], and Ladislava Řezníčková[1,2]

[1]Institute of Geography, Masaryk University, Brno, Czech Republic
[2]Global Change Research Institute, Czech Academy of Sciences, Brno, Czech Republic
[3]Institute for Hydraulic Engineering and Water Resources Management, Vienna University of Technology, Vienna, Austria
[4]Department of Historical Auxiliary Sciences, Institute of History, University of Szeged, Hungary
[5]Department of Geography, Justus Liebig University, Giessen, Germany
[6]Centre for International Development and Environmental Research, Justus Liebig University Giessen, Giessen, Germany
[7]School of Environment and Technology, University of Brighton, Brighton, United Kingdom
[8]School of Geography, Archaeology and Environmental Studies, University of the Witwatersrand, Johannesburg, South Africa

*Correspondence to*: Rudolf Brázdil (brazdil@sci.muni.cz)

**Abstract.** The use of documentary evidence to investigate past climatic trends and events has become a recognised approach in recent decades. This contribution presents the state of the art in its application to droughts. The range of documentary evidence is
very wide, including: general annals, chronicles, and memoirs, diaries kept by missionaries, travellers and those specifically interested in the weather, the records kept by administrators tasked with keeping accounts and other financial and economic records, legal-administrative evidence, religious sources, letters, marketplace and shopkeepers' songs, newspapers and journals, pictographic evidence, chronograms, epigraphic evidence, early instrumental observations, society commentaries, compilations and books, and historical-climatological databases. These come from many parts of the world. This variety of documentary information
is evaluated with respect to the reconstruction of hydroclimatic conditions (precipitation, drought frequency and drought indices). Documentary-based drought reconstructions are then addressed in terms of long-term spatio-temporal fluctuations, major drought events, relationships with external forcing and large-scale climate drivers, socio-economic impacts and human responses. Documentary-based drought series are also discussed from the viewpoint of spatio-temporal variability for certain continents, and their employment together with hydroclimate reconstructions from other proxies (in particular tree-rings) is discussed. Finally,
conclusions are drawn and challenges for the future use of documentary evidence in the study of droughts are presented.

## 1 Introduction

The term "drought" encompasses a complex phenomenon; it is used to express a prolonged period of negative deviation in water balance compared to the climatological norm in a given area (Wilhite and Pulwarty, 2018). Drought is a normal, recurrent feature of climate that occurs in virtually all climate zones (Svoboda and Fuchs, 2018). One of the related environmental phenomena
associated with drought is desertification, which not only has an impact on the environment but may also have severe consequences for human society (e.g. Trnka et al., 2018). Thus a better understanding of the processes leading up to droughts, including their predictability, is highly relevant to social well-being and individual quality of life.

Heim (2002) divided droughts into four categories: a) meteorological, b) agricultural, c) hydrological and d) socio-economic. Fig. 1 shows the sequence of drought occurrence and impacts for generally-accepted types. Mishra and Singh (2010)
add underground water drought to these. Recently, Van Loon et al. (2016a, 2016b) proposed reworking the concept of drought to include the human role in mitigating and enhancing drought. The current authors consider drought the result of "*complex*



*interactions between meteorological anomalies, land surface processes, and human inflows, outflows, and* [water] *storage changes*".

Drought may be defined in terms of several different characteristics and indices (e.g. Heim, 2000, 2002; Keyantash and Dracup, 2002; Svoboda and Fuchs, 2018). SPI (Standardized Precipitation Index; McKee et al., 1993) is one of those frequently
employed in examinations of meteorological drought. SPI is based on measured precipitation, intrinsically excluding all other processes that affect water balance, such as evaporation, run-off, and mean changes in soil-water content. Alternative indices have therefore been proposed. For example, SPEI (Standardized Precipitation-Evapotranspiration Index) requires precipitation and evapotranspiration data (Vicente-Serrano et al., 2010; Beguería et al., 2014) and the Palmer Drought Severity Index (PDSI) includes meteorological as well as soil-related information (Palmer, 1965). Despite such a variety of indices, there is specialist
consensus that no particular one "best" defines drought, and some may be more appropriate than others depending on the investigation in question (Van Loon, 2015). Raible et al. (2017) tested water balance models (differing in the number of hydrological fluxes included) to define drought indices of diverse complexity for several regions in Europe. Their comparison provides insight into regions where indices with simpler water balance models (i.e. a reduced number of hydrological fluxes included) are sufficient to characterize a drought. They demonstrated that SPI, the simplest index, performs well for Western and
Central Europe.

A range of indices have been used in many related studies analysing world-wide drought patterns on the basis of temperature and precipitation measurements in the instrumental period (e.g. Briffa et al., 2009; Dai, 2011; Sousa et al., 2011; van der Schrier et al., 2013; Spinoni et al., 2014, 2015), sometimes covering a time-span of two or three centuries (e.g. van der Schrier et al., 2007; Todd et al., 2013; Haslinger and Blöschl, 2017). For the pre-instrumental period, two data sources in particular may
provide drought information at quite high temporal resolution: tree-rings and documentary data. The growth of trees, reflected particularly in tree-ring widths, late-wood density and isotopes, may react significantly to moisture availability. Using long-term tree-ring series to reconstruct hydroclimate variables constitutes an important part of dendroclimatology (e.g. Hughes et al., 2011). Such reconstructions may reflect spring-summer precipitation (e.g., Brázdil et al., 2002; Oberhuber and Kofler, 2002; Touchan et al., 2005; Wilson et al., 2005, 2012; Kern et al., 2009; Cooper et al., 2012; Dobrovolný et al., 2018) or indicate drought directly, as
expressed by PDSI (Esper et al., 2007; Nicault et al., 2008; Büntgen et al., 2010a, 2010b; Touchan et al., 2010; Cook et al., 2014). In a recent approach, Cook et al. (2015) presented the Old World Drought Atlas (OWDA), a new, tree-ring-based reconstruction of the summer season (JJA) self-calibrated Palmer Drought Severity Index (scPDSI; van der Schrier et al., 2013). For a still more recent world-wide overview of hydroclimate reconstructions based on tree-rings and other natural proxies, the reader is referred to PAGES Hydro2k Consortium (2017); for African hydroclimatic variability over the past 2000 years see Nash et al. (2016a).

While tree-ring information may facilitate resolution of precipitation or drought reconstruction for a specific season of the year, such seasonal restrictions do not apply to documentary evidence (e.g. Brázdil et al., 2005, 2010). Depending on the spatio-temporal availability of weather-related documentary data, it is possible to reconstruct monthly, seasonal and annual precipitation indices (e.g. Glaser et al., 1999; Pfister, 1999; Glaser, 2001, 2008), precipitation totals (e.g. Pfister and Brázdil, 1999; Dobrovolný et al., 2015a), individual droughts (e.g. Dodds et al., 2009; Hao et al., 2010; Zhang and Liang, 2010; Brázdil et al., 2013; Wetter at
al., 2014; Brázdil and Trnka, 2015; Kiss and Nikolić, 2015; Kiss, 2017), drought frequency (e.g. Barriendos, 1997; Piervitali and Colacino, 2011; Brázdil et al., 2013; Noone et al., 2017) and drought indices (Brázdil et al., 2016a; Možný et al., 2016b).

The aim of this article is to present the state of the art for spatio-temporal analyses of droughts derived from documentary evidence for a special issue of *the Climate of the Past* entitled "Droughts over the centuries: What can documentary evidence tell us about drought variability, severity and human responses." Sect. 2 describes the types of documentary evidence that contain
drought-related information. The general features of such evidence, together with reconstructions of precipitation series, drought



frequency series and drought indices series are discussed in Sect. 3. Sect. 4 provides an overview of papers on documentary-based droughts with respect to their long-term fluctuations, severity, the effects of atmospheric circulation and forcings, socio-economic impacts and human responses. Spatio-temporal drought variability in a number of areas and the use of documentary-based droughts in hydroclimate reconstructions from other natural proxies are discussed in Sect. 5. This is followed by some concluding remarks

and perspectives upon future drought studies in Sect. 6.

**2 Documentary evidence**

Detailed descriptions of documentary evidence related to past droughts available in a range of countries may be found in, for example, Nicholson et al. (2012a), Brázdil et al. (2013), Brázdil and Trnka (2015), and Domínguez-Castro and García-Herrera (2016). The main source types of documentary data used for such investigations are described below. This overview provides

examples of them. It does not aim to provide an overview of spatial world coverage, since practically all the types of documentary evidence are potentially available in any part of the inhabited world with a written tradition, whether or not utilised in drought reconstructions to date. Typical examples have been chosen to describe each (commonly known and defined) source type in a brief and informative way; the length of each section is not proportionate to the spatial distribution of available drought-related documentary sources or published analyses.

**2.1 Annals, chronicles, memoirs, inscriptions**

Annals, chronicles, memoirs and inscriptions are narrative sources that report weather/climatic anomalies, including droughts, in general terms. As well as describing drought impacts, such sources often contain specific information about periods without rain. For example, the emperor's annals of the Shang dynasty in China (Emperor T'hang) mention a great drought for 1765 BC, continuing for seven years until 1759 BC (Medhurst, 1846, p. 351): Anno 32: "*This year there was a great drought.* "; Anno 33,

T'hang 19: "*A great drought.*"; Anno 34, T'hang 20: "*The great drought continued.*"; Anno 35, T'hang 21: "*Still a great drought, when the Emperor gave forth the gold of Schwang-san to be coined into money, for the relief of the people*."; Anno 36, T'hang 22: "*The same drought.*"; Anno 37, T'hang 23: "*Still drought.*"; Anno 38, T'hang 24: "*The drought as before. The emperor prayed in the mulberry grove, blamed himself for six things, when the rain fell.*" On the other side of the world, a drought record from the Regensburg annals (Germany) reports drought in AD 823 (Pertz, 1826, p. 93): "*Hard winter, equally severe drought, and great*

*famine.*" A chronicle by Gregorio Susanna from the Calabrian region (Italy) also reports drought impacts in his record for 31 December 1760 (Diodato and Bellocchi, 2011, p. 192): "*Food supplies have been very low because of the great drought that never seems to stop. Decimating all fruits, with grapes also destined to perish, and very little must and wheat and oil* […] *Drought has occurred because there has been no rain up to late December, the countryside is arid and bare of grass, and almost all the cattle are dead. Starvation threatens; much prayer is in order.* […]."

**2.2 Diaries**

For the purposes of this contribution, diaries are considered more-or-less regular daily visual observations of the weather and related phenomena, recorded by their authors for various reasons in ephemeries, calendars, and work-related or personal diaries (for a review of private diaries see e.g. Adamson, 2015). Some of these concentrate largely upon weather records like that, for example, kept (in Latin) by the Reverend William Merle, Rector of Driby, Lincolnshire, England, between 1337 and 1344. In his

record for May 1342 he mentions a drought that year (Symons, 1891, unpaginated): "*It is to be noted that not much rain fell between the 9th of April and 20th of May, but there was dryness* [siccitas] *with considerable heat.*" He returned to this drought in



his final comment for 1342 (ibid.): "*It is to be noted that there was dryness about April this year, as there was in the years of Our Lord 1333 and 1340.*"

Detailed daily weather records may also be gleaned from standard diaries from among other, non-meteorological, information – see, for example, the diaries of the Premonstratensian order in Hradisko monastery and abbey Svatý Kopeček near Olomouc (Czech Republic), which span the 52 years from 1693 to 1783 (Brázdil et al., 2008a). Drought is reported in these diaries for 17 years, with particular mention of those that occurred during the April to September period so critical to agricultural crop production. For example, according to the record for 19 September 1750, drought prevented autumn sowing, and prayers for rain was organised on 23 September (archival source 7 – AS7).

Missionary and travel diaries are also important sources of weather information for regions in which meteorological observations started comparatively very recently. One example is provided by Peter Heinrich Brincker, a German missionary of the Rheinische Missionsgesellschaft from Wuppertal, who described a drought in Otjikango, central Namibia, in a report dated 10 September 1869 (AS1; Grab and Zumthurm, 2018, p. e656): "*The drought and its consequences, the hunger, impacts us all very hard and many poor people are dying from exhaustion. Yes, some days back it was mentioned that amongst the Ovatjimba or poor Herero, the hunger is so large, that they have resorted to cannibalism, which might well be an exaggeration.*"

**2.3 Financial and economic-administrative records**

Financial and economic-administrative sources consist of documents prepared at various levels of governmental or state administration. For example, damage to agricultural production related to extreme hydrological and meteorological events constituted grounds, in some regions, for the rebate of taxes, a system well-developed in the Czech Lands in the 17th–20th centuries (e.g., Brázdil et al., 2006, 2012; Dolák et al., 2013). A plethora of documents relevant to Czech application process are preserved at the levels of local, regional and state administration. A good example for this important source type may be cited in the form of a request for financial support from the Valašská Polanka settlement, addressed to the I. R. District Office in Valašské Meziříčí on 28 August 1904 (AS6): "[...] *this year* [1904]*, our poor peasants are very distressed, because the grain – due to great drought – became dry 14 days earlier than in other years, the grains are small and the straw on local hills has remained short,* [there was] *no aftermath from dried-out meadows and no second clover,* [while the] *potatoes in some places* [are] *already dry and small and starting to rot, and of cabbage there is nearly none. Because there is neither aftermath nor second clover,* [people] *are having to give their livestock dry feed already, of which there is very little, and therefore, for shortage of feed, much livestock, the main farming resource in the local mountainous landscape, has to be sold off by whatever means* [...]". An example from Asia may be found in an imperial edict from the early spring of 1878 in Shangyudang, reporting a dry episode in northern China (Hao et al., 2010, p. 2003): "*Shanxi, Henan, Shaanxi and other provinces in the prior year* [1877] *experienced severe drought, snow in the winter* [1877/1878] *was insufficient; although the farmers prayed for many times, it did not get enough precipitation yet.*"

**2.4 Legal-administrative evidence**

Legal trials related have helped preserve the memory of some significant drought events and water scarcity problems, sometimes in indirect and curious fashion. For example, the infamous witch trials of the past often arose out of the need to blame human or supernatural agencies for natural misfortunes; in such cases the men or women accused of witchcraft were often held responsible for drought by their "taking the rain/dew away." A typical example is the great Szeged (Hungary) witchcraft trial that started in 1718 and ended in 1728 and 1729 (see e.g. Petrovics, 2005). Drought was reported in the witchcraft trial confession protocols in 1728 (Reizner, 1899, pp. 375–376), as follows: "*10th witness. The proud and respected Stephanus Vastagh, citizen, aged c. 80. Confessed in Christian faith that not long ago the problem of the great drought was mentioned, and also that it would be in fact a*





*scourge of a drought. Pál Kovács, the judge of the beggars, suggested that this great drought was not from God, but in fact the rain was sold to Turkey by the witches.*" There also exists legal/administrative evidence in the form of a juridical manuscript from Litoměřice (Czech Republic) that contains a record dated to 1503 (AS8): "*The summer was so dry that such aridity could not be recalled in thirty years* [and] *on many brooks and rivers they could not mill because of the drought, and there was a bad harvest in*

*the fields; of the spring grain there was almost nothing and in many places they had to tear it off* [by hand] *because it was impossible to reap; the wine was very good this year and* [there was] *great frost this summer in the mountains on St. Laurentius for three full days* [20–22 August], *but the dryness was huge* [sic]." Other controversies generated by drought could be related to the use and allocation of water resources, perennial problems that always tend to be exacerbated by periods of hot and dry weather (see e.g. Kiss, 2017). A third type of legal documentation relates to disputes centring around field surveys, land ownership and the

boundaries of lands that were made available for utilisation in multi-annual dry/drought periods. For example, along the southern shoreline of Lake Fertő/Neusiedl in western Hungary, land surveys, documented investigations of ownership debates over dried pastures in detail, together with descriptions of 17th–18th-century drought events presented by eye-witnesses (Kiss, 2005).

### 2.5 Religious sources

Rogation ceremonies – formal, organised congregations and processions gathered to beseech God for rain – are described in

important ecclesiastic sources. They tend to be a feature of Roman Catholic church accounts, but were also reported by other branches of religion. Rogation ceremonies were performed for two main reasons: to plead with God for the end of drought (*pro-pluvia* rogations) or cessation of wet/stormy periods (*pro-serenitate* rogations) that endangered or even damaged agricultural production (e.g. agricultural drought). They were often included in "Chapter Acts", i.e. documents produced by local governments. *Pro-pluvia* rogations provide a good drought indicator from the 15th to the 19th centuries in Europe (in certain areas in Spain they

are performed to this day). For example, a report for level I of the rogation ceremony in Barcelona from 3 March 1691 reads (AS2; Martín-Vide and Barriendos Vallvé, 1995, pp. 217–218): "*The syndic of the city* [municipal authority] *has arrived in the Chapter* [ecclesiastical authority] *on behalf of the Council, bringing the message of the very great need for Water and they have deemed it justified that the Chapter convoke rogation ceremonies; after hearing this message, the Chapter has determined to follow the usual way in similar situations, and to have the prayer for rain at Mass, beginning on the 4th of this month* [March] […]". Martín-Vide

and Barriendos Vallvé (1995) and Barriendos (1997) developed a Drought Rogation Index, taking into account five levels of drought intensity/duration according to the hierarchic system of rogation ceremonies performed in Barcelona (Spain). Rogation ceremonies as dry/wet proxies have been analysed in many papers covering Spain and Portugal (see Domínguez-Castro et al., 2008, 2012; Fragoso et al., 2018; Tejedor et al., 2018 and references therein), Italy (Piervitali and Colacino, 2001) and France (Garnier, 2014). Rogations are also well-known in former Spanish colonies in Central and South America (e.g. Domínguez-Castro

and García-Herrera, 2016; Domínguez-Castro et al., 2018; Guevara-Murua et al., 2018); in Mexico starting in the 1590s (Garza and Barriendos, 1998; Garza Merodio, 2002); in Equador, Peru and Argentina since the 16th century (Huertas Vallejos and Ortlieb, 1992; Prieto et al., 2000; Prieto and García Herrera, 2001, 2009; Gascón and Caviedes, 2013); in Colombia between the 16th and 20th centuries (Jurado Jurado, 2004; Ulloa, 2014); and in Brazil from the 18th century to the current day (Heathcote, 2013). Finally, rogations have also been reported from the Philippines since the 16th century (Warren, 2013).

### 2.6 Letters

Letters of a private or official character may include direct information about drought, activities influenced by drought, and other drought impacts or consequences. For example, Petr Uher, an officer in Uherský Brod (Czech Republic), wrote in a letter to Commander Albrecht Václav Eusebius of Waldstein on 3 November 1626, during the Thirty Years War (1618–1648) in Europe,



responding to a request concerning the baking of bread for his forces. He noted low water levels in rivers due to an ongoing drought that had affected the operation of watermills. This was obstructing the supply and milling of grain (Hrubý, 1937). On 4 March 1791, Governor Phillip wrote from the recently established British colony of New South Wales (Australia), providing information on drought to the Right Honourable W.W. Grenville (Nicholls, 1988, p. 4): "*From June* [1790] *until the present time*

*so little rain has fallen that most of the runs of water in the different parts of the harbour have been dried up for several months, and the run which supplies this settlement is greatly reduced, but still sufficient for all culinary purposes* [...] *I do not think it probable that so dry a season often occurs. Our crops of corn have suffered greatly from the dry weather*."

### 2.7 Marketplace and shopkeepers' songs

Marketplace and shopkeepers' songs provide an expression of certain extreme events in artistic or common form, describing both
conditions at the time (e.g. flood, windstorm, hail) and their damaging effects. Two examples have been described from the Czech Lands (Brázdil and Trnka, 2015; Fig. 2). The severe drought in 1678 was the inspiration for the song "Key to the Rain, or a New Song for a Time of Drought" (*Klíč od deště aneb Nová píseň v čas sucha*), published in Prague in 1678 and a year later (AS4). From its first lines it follows that the first post-drought rain fell as late as Saint Wenceslas' Day (i.e. 28 September): "*In this year of grace one thousand / And at the same time six hundred / And seventy-eight / On the Day of Saint Wenceslas / Honour and glory be*
*to God / Water is given to us from the sky. / At the intercession of the Mother of God /* [Crop] *Yields gladly increase / Little rains serve well. / Through heaven's Majesty / A little rain fell in quietness / Aside from all human concerns*." A similar song entitled "Song in Need of Rain" (*Píseň za déšť potřebný*), related to a severe drought in 1790, stems from the manuscript of Antonín Štěpán, a wealthy citizen of Pelhřimov (Martínková, 2005).

### 2.8 Newspapers and journals

Droughts and their impacts (e.g. lack of water, harvest failure, increases in prices, famine) are types of hydrometeorological extreme frequently reported in newspapers and magazines. The validity and importance of such sources for use in historical-climatological studies have been pointed out in many papers (e.g., Gallego et al., 2008; Munro and Fowler, 2014; Nash et al., 2016b; Noone et al., 2017). For example, the 10th volume of *Theatrum Europaeum* (a printed work by Matthäus Merian related to the history of German-speaking countries, published in 21 volumes between 1633 and 1738) reported that during the hot summer
of 1666, meadows and streams in Austria, Bohemia and Hungary dried up, people had to go 6–7 miles [*c.* 45–53 km] for water, and fires broke out in villages and forests. An example of a newspaper report (*Moravské noviny*, 17 June 1885, No. 138, non-paginated) mentions drought and associated impacts on the yields of agricultural crops from Pozořice and surroundings (Czech Republic): "*The dry weather has already reached such a level that nearly everything has started to dry up. The rye, wheat, oats and early barley are staying strong, although the maize is developing very slowly. The late barley, if there is no rain by Sunday* [21
June 1885]*, will be beyond recall, since it has already turned yellow and dried up completely. The latest* [barley] *is not even coming into ear. The legumes are coming to the end of their flowering very poorly and are also about to dry up*. [...] *The fruit appeared to be coming on nicely; now it has already started to fall, Walnut trees have suffered the most. The clover was sparse* [...]." Farther afield, another example of newspaper coverage reports a drought (1887–1888) in Tasmania (*Mercury*, 3 February, 1888, p. 3; Evans, 2012, p. 209): "*The weather continues to be dry and hot, and the surface of the earth* […] *presents a uniformity*
*of parched and blackened barrenness. The remains of what was once grass is now almost powder, and could be blown away, and the young growth of trees, which would otherwise yield a subsistence for cattle, has been consumed by the bush fires.*"

### 2.9 Paintings and pictographic evidence



Paintings and pictures have been used throughout history to present and commemorate various events, including those related to the weather, that have had important impacts upon societies. The Aztec people in central Mexico produced pictorial "books" (or "codices") painted on skins or native *amatl* (bark) paper, which described religious and political events and natural phenomena for individual city-states (*alteptl*). Therrell et al. (2004) identified 13 drought years between AD 1332 and 1543 from such codices.

The Aztec belief was manifested in cyclical drought-induced famines associated with the calendar icon known as One Rabbit (Fig. 3). Nine of the identified droughts were also reflected in below-average tree-ring widths in Douglas fir chronologies from central and northern Mexico (ibid.).

Gallo and Wood (2015) used Native American pictographic evidence (known as "winter counts") for the study of past drought events in the U.S. Great Plains. Some Native Americans recorded the major events of the year, defined generally as the

period between the first snowfalls of successive years. Winter counts were the responsibility of a specific keeper who would consult with a council of elders as to which events were important enough to be recorded. From these counts, Gallo and Wood (2015) identified ten prolonged dry or drought events between 1700 and 1880 that correspond with other observations or available information (particularly with PDSI values from Cook and Krusic, http://iridl.ldeo.columbia.edu/SOURCES/.LDEO/.TRL/.NADA2004/.pdsi-atlas.html, last access: 27 August 2018).

**2.10 Chronograms**

Chronograms are a form of record commemorating any important event, including weather anomalies. Selected letters in the record are then interpreted as Roman figures (in the form of large or highlighted letters) and indicate the year of the event. One example is a Latin entry in the records of Hieronymus Haura, a member of the Augustinian order in Brno (Czech Republic), commemorating the drought of 1746 (AS5):

"Personat he**V**! tu**I**st**I**s **V**o**X**: **SVCCI**s aret a**D**e**M**pt**I**s / Noster ager s**I**t**IV**nt fontes, herbaeq**V**e, feraeq**V**e."

In translation, this reads: "*It resounds! Oh, woe betide, such a voice: The drought desiccated, it eats up our fields, the springs are thirsty, the plants and animals, too.*" The year 1746 is then given as the sum of the highlighted Roman figures, i.e., V + I + I + V + X + V + C + C + I + D + M + I + I + I + V + V + V (5 + 1 + 1 + 5 + 10 + 5 + 100 + 100 + 1 + 500 + 1000 + 1 + 1 + 1 + 5 + 5 + 5 = 1746) (Brázdil and Trnka, 2015).

**2.11 Epigraphic evidence**

Low water levels in rivers were often recorded by marks on large stones located in river beds and appearing during low water stages, in this way identifying long periods of hydrological drought. Because drought periods were usually accompanied by bad harvests and even famines, they became known in some areas as "hunger stones". Using this type of epigraphic evidence requires analysis of the reliability of the marks in question, as some of them could have been created without direct relation to a low-flow

event. A stone located on the left bank of the River Elbe at Děčín-Podmokly (Czech Republic) is one example (for other hunger stones on the Elbe in German Magdeburg and Dresden see Elleder, 2016). Records chiselled on the stone commemorate low water levels in 1417, 1616, 1707, 1746, 1790, 1800, 1811, 1830, 1842, 1868, 1892, and 1893 (Fig. 4). They are accompanied by the inscription: *Wenn du mich siehst, dann weine* (If you see me, then weep), warning of the consequences of drought, e.g. a bad harvest, lack of food, high prices and hunger for poor people. A similar hunger stone known as the "Laufenstein" was used by

Pfister et al. (2006) to study low water levels indicating hydrological drought in the River Rhine in Switzerland; a stone named the "dearth stone" is also known on the River Danube at Budapest (Hungary) (Palotay et al., 2012).

**2.12 Early instrumental observations**



Instrumental meteorological observations, initiated by individuals or official bodies before the establishment of standard meteorological networks in certain countries, may be considered "early instrumental observations" (e.g. Camuffo and Jones, 2002; Domínguez-Castro et al., 2014b, 2017; Slonosky, 2014). As well as instrumental data, they may also incorporate weather descriptions, including hydrometeorological extremes such as drought. For example, Thomas Heberden added information about

very dry years in 1749 and 1750 to his meteorological measurements (1747–1753) at Funchal (Madeira), in which "*the corn was destroyed, and the fruit-trees suffered much, particularly the peach-trees, the fruit either falling to the ground, whilst green, or, if it remained longer on the tree, being full of white worms*" (Alcoforado et al., 2012, pp. 357–358).

## 2.13 Society reports

Information related to drought impacts also appears in the reports/publications of various learned and other societies engaged in
agricultural and forestry production, particularly in the 19th century. For example, the I. R. Patriotic-Economic Society in Bohemia, established to enhance economic development in the country, published annual overviews of agricultural production for 1822–1845, supplemented after 1828 by forestry management reports. At the same time, the Society organised a network of meteorological and phenological stations, together with publications of the measurements taken by them (see e.g. Brázdil et al., 2011; Bělínová and Brázdil, 2012). For example, the drought of 1835 in Bohemia was reported as follows (Neue Schriften, 1837,
p. 196): "*The year 1835 was characterised, just like the previous year, by unusually long-lasting drought and high temperatures in the lowlands, followed in many places by weak yields of grain and particularly of feedstuff, while in the mid-mountainous regions the required moisture was not absent, and so a rich yield of the majority of agricultural products followed in them. Lack of rain was particularly marked and led to crop failure in the immediate surroundings of the capital* [Prague] *and* [also] *largely in the* [České] *Budějovice, Plzeň and Žatec districts.*"

## 2.14 Weather compilations

Weather compilations, including information about drought events, are available for many different areas and time intervals (e.g. see Weikinn, 1958–2002 for Europe; Réthly, 1962, 1970, Réthly and Simon, 1999 for the Carpathian Basin; for a critical source evaluation see Kiss, 2009). If such compilations are not prepared by historians, past and present, biases may arise out of unreliable documentary sources that leads in turn to the inclusion of erroneous data and the duplication or misidentification of reported events
(e.g. Bell and Ogilvie, 1978). Such compilations should, therefore, be checked for socio-temporal reliability, and any unreliable, uncertain, and clearly non-contemporary references removed. Even better source collections or databases, compiled by skilled historians with a source-critical approach (e.g. Malewicz, 1980; Alexandre, 1987), sometimes require additional source analysis. Despite these problems, the importance of such compilations or books should be emphasised here, as they may be the only available sources for direct climatological analyses in certain regions. For example, Borisenkov and Pasetskiy (1988) published a
thousand-year history of unusual natural events, including drought events (*zasucha*), based on Russian written sources (for an extension of unusual natural events back to the 5th century BC see Borisenkov and Pasetskiy, 2002). The same study also reported droughts beyond Russian territory, particularly in "western Europe".

## 2.15 Historical-climatological databases

Several historical climatology research groups have developed databases containing information about drought and related
phenomena (Decker, 2018). The first such database, with roots in the late 1970s, was initiated by the historian C. Pfister in Bern, containing data not only from his native Switzerland, but also for other countries within Europe and beyond. The historical





development of the Euro-Climhist database and its recent status are described in detail on http://www.euroclimhist.unibe.ch (last access: 27 August 2018). In Germany, R. Glaser (University of Freiburg) was a prime mover in the creation of the HISKLID historical climatological database, which was subsequently extended into CRE tambora.org, a new web-based Collaborative Research Environment (CRE) for historical climatology and environmental research (Riemann et al., 2015). Documentary data and early instrumental meteorological observations for Japan are available at http://jcdp.jp (27 August 2018), a result of rescue activities under the Japan-Asia Climate Data Program. A large set of Chinese documentary sources, published by D. A. Zhang in 2004 (first edition) and 2013 (second edition), was digitised under the supervision of P. Wang from the Research Center for Environmental Changes, Academia Sinica, Taipei, in the form of the REACHES (Reconstructed East Asian Climate Historical Encoded Series) database (Wang et al., 2018).

## 3 Methods of drought reconstruction from documentary evidence

The individual categories of drought reported in Sect. 1 may be well described by the various types of documentary data mentioned in Sect. 2. As follows from Table 1, reliably-dated accounts describing a lack of precipitation, as well as information summarising dry periods during particular months or seasons, can be used to identify meteorological droughts. Accounts of low water levels in rivers, lakes and ponds, together with descriptions of a general shortage of water, are clear signals of the occurrence of hydrological drought. Bad or failed harvests of agricultural crops, accompanied by a description of their causes, can be indicators of agricultural drought. Because documentary data also provides important details about societal behaviour, accounts of drought consequences and impacts can be used to detect socio-economic drought.

The extraction of drought information from documentary sources requires a source-critical approach (e.g. see Brázdil et al., 2005, 2010). The use of primary sources and careful cross-checking of data are essential to avoid erroneous interpretation of documentary evidence. Careful analysis and interpretation of evidence is also required to identify the true spatial extent, duration, severity and impacts of individual drought events.

A number of approaches may be taken to the compilation of long-term series of drought frequency and/or magnitude from documentary evidence. Rich documentary evidence allows the creation of various types of precipitation/drought indices, many of which may result in different hydroclimate reconstructions:

**(i) precipitation series**

Various precipitation indices may be developed, depending on the availability and quality of documentary evidence. The precipitation character of months/seasons/years may be categorised by various scales. For instance, 3-degree (−1 − dry, 0 − normal, 1 − wet), 5-degree (−2 − very dry/drought, −1 − relatively dry, 0 − normal, 1 − relatively wet, 2 − extremely wet) or 7-degree (−3 − extremely dry, −2 − very dry, −1 − dry, 0 − normal, 1 − wet, 2 − very wet, 3 − extremely wet) scales are the most widely used in Europe (e.g. Pfister, 1992, 1999, 2001; Glaser, 2001, 2008; Dobrovolný et al., 2015a) and Africa (e.g. Nicholson et al., 2012a, 2012b; Nash et al., 2016b, 2018). Monthly indices may be added up to obtain seasonal (e.g. between −9 and 9 in a 7-degree scale) or annual (e.g. between −36 and 36 in a 7-degree scale) series of indices (Fig. 5a). Other similar scales may also be used (*cf.* Rodrigo et al., 1999; van Engelen et al., 2009; Ge et al., 2016; Guevara-Murua et al., 2018). Where there is an overlap of indices with series employing instrumental precipitation records, standard palaeoclimatological approaches based on the use of calibration/verification periods may be employed for quantitative precipitation reconstruction (see e.g. Dobrovolný et al., 2009, 2015a). Any lack of an overlapping period between documentary and instrumental data is more challenging. However, Rodrigo (2008) presented one approach to reconstructing precipitation for decadal or longer time units based on counting the number of extreme events in the past and inferring mean and standard deviation using the assumption of normal distribution. This approach



was used to develop winter rainfall series for 30-year periods in Andalusia (Spain) for 1501–2000. Quantitative precipitation series may subsequently be used for the statistical analysis of wet and dry periods, trends and extremes in the long-term.

**(ii) drought frequency series**

The types of drought information recorded in documentary evidence allow selected drought episodes to be identified and presented
as long-term chronologies (e.g. Gioda and Prieto, 1999; Jiang et al., 2005; Mendoza et al., 2005a; Noone et al., 2017). In these types of series, the criteria used by individual authors to identify droughts may have significant influence upon homogeneity. For example, Brázdil et al. (2013), applying precipitation indices between –1 and –3 (see point i), used at least two consecutive "dry" months to define episodes of drought for the Czech Lands in the pre-instrumental 1501–1803 period. In the instrumental period from 1804, the frequency of drought episodes was compiled from the calculation of SPEI-1 and Z-index values, from which those
of concurrent occurrence and a return period of two years or more were reflected as drought episodes to obtain a long-term drought chronology (Fig. 5b).

**(iii) drought index series**

Documentary-based precipitation and temperature reconstructions may be used to obtain main drought indices from documentary data. For example, Brázdil et al. (2016a) calculated seasonal, summer half-year and annual SPI, SPEI, Z-index and PDSI series for
the Czech Lands (Fig. 5c), using a monthly temperature reconstruction for central Europe (Dobrovolný et al., 2010) and a seasonal precipitation reconstruction for the Czech Lands (Dobrovolný et al., 2015a), both starting in AD 1501. However, calculation of drought indices from documentary data is still in its early stages.

    A further drought index reconstruction from the Czech Lands has been derived from series of grape harvest dates, which are normally used only for air temperature reconstructions (e.g. Chuine et al., 2004; Meier et al., 2007; Maurer et al., 2009; Moreno
et al., 2016). The series created for the Bohemian wine-growing region (particularly north-west of Prague) since AD 1499 has been used not only for the reconstruction of April–August temperatures (Možný et al., 2016a), but also for the reconstruction of April–August SPEI (Možný et al., 2016b). This SPEI reconstruction (Fig. 5d) is a challenge for testing similar relationships and possible drought reconstructions in other relatively drier wine-growing areas in Europe.

**4 Results**

This section presents a global overview of papers dealing with documentary-based drought reconstructions with regard to long-term spatio-temporal variability, major extreme events, atmospheric circulation and climate forcing, socio-economic impacts and human responses.

**4.1 Long-term precipitation and drought series**

As a large number of papers describing documentary-based droughts exist worldwide, this section is further divided
geographically, by individual continents.

**4.1.1 Europe**

In the British Isles, Ogilvie and Farmer (1997) used "precipitation scores" to classify the character of monthly precipitation for England in AD 1200–1439 based on available documentary evidence. For dry patterns they used scores of –2 (slightly more dry than normal) and –3 (particularly dry). Despite many absent scores, it was possible to interpret precipitation character for June and
July, at the least. Several papers have addressed drought conditions in Ireland. Noone et al. (2017) used SPI values calculated from a precipitation network (1850–2015) and reconstructed precipitation from newspapers and other documentary data from 1765 to



create a drought catalogue for the island of Ireland. Exceptionally long drought periods were identified for 1854–1860 (continuous droughts) and 1800–1809 (three droughts with brief interludes). For the period after 1850, they reported six other droughts with island-wide fingerprints and increasing dryness expressed by SPI from the 1990s onwards. Murphy et al. (2017) compiled a 250-year drought catalogue for Ireland. Although most of the descriptions concentrated on particular events and their socio-economic

consequences, drought poems as well as prayers for rain were also found among newspaper entries. Multi-seasonal droughts occurred in the 1800s, 1820s, 1850s, 1880s, 1920s, 1930s, 1950s and 1970s (some of the characteristic drought years, such as 1806, 1887 and 1893 were analysed individually). More recently, Murphy et al. (2018) extended the Ireland monthly precipitation series back to 1711 using early instrumental and documentary series from before 1850.

For central Europe, Gimmi et al. (2007) used eight different weather diaries from observers around the city of Bern

(Switzerland), to compile qualitative information about precipitation from 1760 to 1863. This data was combined with precipitation measurements from 1864 onwards to reconstruct precipitation series for Bern for the entire 1760–2003 period. As noted in Sect. 3, several papers have reported on droughts in the Czech Lands. Brázdil et al. (2013) analysed droughts for the past millennium based on documentary data and instrumental records. Although the oldest credible documentary source speaks of a dry winter in AD 1090/1091, only 36 drought episodes prior to AD 1501 were reported in the Czech documentary evidence. Based on monthly

drought indices in the pre-instrumental period between 1501 and 1803, they delimited various drought episodes; these were combined with SPEI-1 and Z-index indices calculated for the instrumental period from 1804 to compile a 500-year drought frequency series (Fig. 5b). Later, Brázdil et al. (2016a) reconstructed series of seasonal, half-year and annual SPI, SPEI, Z-index and PDSI for the 1501–2014 period (Fig. 5c). The driest episodes occurred around the beginning and end of the 18th century, and 1540 was identified as a particularly dry-extreme year. Možný et al. (2016b) reconstructed April–August SPEI series from grape

harvest dates in the Bohemian hop- and wine-growing region north-west of Prague for AD 1499–2012 (Fig. 5d). The reconstructed SPEI series, explaining 75% of the drought variability since 1841, showed a firm agreement with other SPEI reconstructions from the Czech Lands based on documentary and instrumental data.

A number of drought series derived from documentary evidence have been developed for Spain, including several papers based primarily on rogation ceremonies. For example, Martín-Vide and Barriendos Vallvé (1995) described in detail the use of

*pro-pluvia* rogation ceremonies, and presented an index of frequency of these rogations according their five defined levels during the 15th–19th centuries for Catalonia (Spain); this was supplemented by fluctuations in flood and drought indices. Subsequently, Barriendos (1997) used data from rogation ceremonies held in several places in Catalonia, Toledo and Sevilla for the creation of a Drought Rogation Index for the 17th–18th centuries. Barriendos (2005) presented detailed liturgical evidence with respect to extreme climatic events. The greater part of his study concerned rogation ceremonies and their employment in the reconstruction of

rainfall variability. He presented a weighted index chronology of droughts for the 1500–1900 period and delimited four episodes of extreme drought: 1562–1568, 1626–1631, 1752–1758 and 1812–1818. Domínguez-Castro et al. (2010) also used rogation series from Bilbao, Catalonia, Zamora, Zaragoza, Toledo, Murcia and Seville to study drought occurrence over the Iberian Peninsula for the 1600–1750 period. While droughts during the first half of the 17th century were quite localised, those in subsequent years affected broader regions, or even the whole peninsula. The two most extended droughts occurred in 1664 and 1680. Tejedor et al.

(2018) compiled a new dataset of rogation ceremonies from 13 cities in the north-east of Spain and investigated the annual drought variability from AD 1650 to 1899. They found common periods with prolonged droughts (during the mid- and late 18th century) and extreme drought years (1691, 1753, 1775, 1798 and 1817) associated with more blocking situations.

In addition to information about rogation ceremonies, Domínguez-Castro et al. (2014a) analysed 11 Islamic chronicles for Iberia, covering the period AD 711–1010 at high temporal and spatial resolution. They identified three severe droughts in 748–754

(drought reported each year), 812–823 (droughts with long famines) and 867–879 (droughts with references to famine). Fernández-




Fernández et al. (2014) described available documentary evidence and qualitative weather records for Zafra, the capital of the Duchy of Feria in the south-western Iberian Peninsula, for the 1750–1840 period. Documentary data, particularly a quasi-weekly report of the weather in Zafra, were used for the reconstruction of a monthly rainfall index, which showed two dry periods, in 1796–1799 and 1816–1819 (Fernández-Fernández et al., 2015).

Rodrigo et al. (1999) used a number of documentary sources from southern Spain to create an ordinary rainfall index and describe monthly patterns since AD 1500 on a 5-degree scale: –2 hydrological drought, –1 weather drought, 0 normal, 1 hard and/or constant rain, 2 rain-induced flood. Results were calibrated with modern precipitation data and other studies of historical climate. The driest periods in the pre-instrumental period were detected in the first half of the 16th century and around 1750. In several other papers reconstructing precipitation from documentary data in Spain (e.g., Rodrigo et al., 1995, 2012; Rodrigo and
Barriendos, 2008), droughts were not directly reported.

For southern Portugal, Do Ó and Roxo (2008) used published and archival documentary evidence, supplemented by newspaper records, to analyse droughts from the 12th to the 18th centuries for the Algarve region (supplemented by data for the neighbouring region of Inner Lower Alentejo). The years 1385–1398 stand out as a long, dry period, while the first halves of both the 16th and the 18th centuries were richer in reported multi-annual drought events. The 17th century was characterized by a low
occurrence of drought reports. In the 19th century, covered mainly by newspapers (referring to long-term dry spells rather than long-lasting drought events), the 1873–1878 drought was identified as the longest documented event with hunger, increased crime, sanitation/disease and other severe problems, culminating around 1875.

For Italy, Piervitali and Colacino (2001) reported 50 prayer processions for the relief of droughts, based on records from Erice in western Sicily relating to the 1565–1915 period. These processions were at their most frequent during the 17th century (21
years, compared to only 8 years in the 19th century) and in April (28 years), followed by May and March (a total of 44 years in any spring month). Later, Diodato (2007) used documentary data for the River Samnium region (continental southern Italy) in the 1675–1868 period to develop a numerical index characterizing the rainfall regime and its evolution. Combining this with the precipitation series of Benevento for 1869–2002, he reconstructed the series for the entire 1675–2002 period. The driest periods were detected in the 18th century and the wettest in the 19th century. In a follow-on paper, Diodato and Bellocchi (2011)
investigated drought conditions in central-southern Italy for 1581–2007, based on documentary evidence. Their drought index (DI) was defined as: value 1 for meteorological drought associated with agricultural drought at least in two places, otherwise DI = 0. A drought year was defined as that with at least three successive months with DI = 1. A drought-weighted index sum (DWIS) was also calculated, taking the duration of the dry episode into account.

Telelis (2008) identified 183 droughts for the eastern Mediterranean and the Middle East using Byzantine documentary
sources for the period AD 300–1500. Taking into account decades with more than two dry events of extended duration, he reported a higher frequency of dry episodes in AD 360–390, 530–580, 690–720 and 1090–1200 for the temperate semi-arid regions, in AD 320–340, 390–420, 450–480, 510–560, 600–630, 740–770, 1040–1070, 1130–1200 and 1290–1320 for the desert region, and in AD 560–590, 740–790, 1020–1050, 1070–1110 and 1140–1160 for the Mediterranean regions. Grove and Conterio (1995) used documentary data to analyse winter and spring droughts, severe winters and summer rains during the period 1548–1648 on the
island of Crete. They concluded that some of the winter droughts were longer-lasting and more extreme than those recorded in the instrumental period, and were probably related to the extension of southerly air masses from the Sahara.

For eastern Europe, Lyakhov (1984) used documentary data to establish the frequency of extremely dry and wet spring–summer seasons for 30-year intervals from the 13th century to 1980 for the European non-chernozem part of the former Soviet Union. Nine extremely dry seasons emerged, in 1351–1380, 1831–1860 and 1891–1920, followed by eight cases in 1201–1230 and





seven in 1411–1440, 1801–1830 and 1951–1980. Only one extremely dry season was identified for both the 1231–1260 and 1771–1800 periods.

### 4.1.2 Asia

In Asia, China provides the most reports of documentary-based droughts. More than 2,200 local chronicles and many other historical writings in China, divided into 120 regions, have been used to create annual maps of dryness/wetness indices for the period AD 1470–1979 (Academy of Chinese Meteorological Science, 1981; for types of Chinese historical documents and climate information see Ge et al., 2008). An example of such a map, for AD 1484, appears in Fig. 6. A 5-degree scale was used for classification: 1 – very wet, 2 – wet, 3 – normal, 4 – dry, and 5 – very dry. The reported dryness/wetness indices were subsequently extended to 1992 by Zhang and Liu (1993) and to 2000 by Zhang et al. (2003), and have been further used as a basic comparative dataset in many Chinese studies, usually analysing floods and droughts together. In the following description of drought analyses in China, we follow Ge et al. (2016) by dividing the Chinese territory into three climatic zones: (i) a monsoonal climate in the east, (ii) a continental arid climate in the north-west, and (iii) a highland cold climate on the Qinghai-Tibet Plateau.

Song (2000), through a statistical analysis of the above dry/wet evidence, reported dry conditions during the growing season from 125 sites in north-eastern China in the 16th and 17th centuries, and prevailing dryness in most areas of China during the 20th century. Qian et al. (2003) used the above data, extended to 1999, to interpret the spatio-temporal variability of dryness/wetness, by means of rotated empirical orthogonal function analysis. They found six principal spatial modes, located mainly in the middle-lower Yellow River valley, the middle-lower Yangtze River valley, north-western China, north-eastern China, southern China and south-western China, but their strength and location fluctuated from century to century. The Yellow River and Yangtze River valleys were identified as the areas with the widest historical dryness/wetness variability in Eastern China. Qian et al. (2012) used the annual dryness/wetness index for AD 1470–2003, combined with instrumentally-recorded precipitation after 1951, to identify extremely dry years and events in the Great Bend of the Yellow River region, near the northern fringe of the East Asian summer monsoon. They detected 49 drought years (an opposite wet pattern prevailed in south-eastern China), of which 26 were severe. Six-year drought occurred in 1528–1533 and four-year drought in 1637–1640. Shorter, very severe, droughts occurred in the 20th century: two-year drought in 1928–1929 and one-year drought in 1900 and 1965. These persistent and extreme droughts caused severe famines and huge loss of human life. Wang et al. (2015), using documentary-based drought data from Eastern China for the period 1470–2000, reported a higher number of droughts during the 16th and 17th centuries than in the 18th and 19th centuries. Droughts were more extreme in these centuries than in the 20th century. They detected 20- and 30-year drought cycles in the periods 1480–1670 and 1825–1940, and ~60-year cycles in the 16th and 17th centuries.

Many Chinese studies have analysed historical droughts and floods together. For example, Chen et al. (2001) assessed the spatial distribution of the ten greatest drought and flood years in China during the previous millennium, presenting descriptions and maps related to each catastrophic year, including the great droughts in 1321 and 1835. Jiang et al. (2005) analysed floods and droughts in the middle Yangtze River and the Yangtze River delta using the frequency of floods and droughts for AD 1000–1950 recorded in collected historical documents, as well as discharges with precipitation totals for later periods. A positive trend in drought frequency was found from AD 1400 to the present. Yin et al. (2005), applying a database from local administrative documentation covering 2300 years, analysed drought hazards along the River Wei, a major tributary of the Yellow River. Two periods between AD 610 and 850 and from 1580 to 2000 (1810–1940 in particular) emerged as major, long-lasting episodes during which droughts and other hydrological hazards were prevalent. Shen et al. (2007) used Chinese drought/flood proxy data for the past 500 years to study exceptional drought events over Eastern China (east of 105°E). They identified three such events in 1586–



1589, 1638–1641, and 1965–1966. Drought occurred over more than 40% of the affected area, with a significant summer rainfall reduction of ~50% or more in the core areas. Yi et al. (2012) combined a documentary-based drought/flood index and tree-ring data for the larger part of north-central China and distinguished between droughts related to the combination of rainless and intensely hot patterns (AD 1484, 1585–1587, 1689–1691, 1784–1786 and 1876–1878), rainless summers driven by low

precipitation and/or high temperatures (1560–1561, 1599–1601, 1609, 1615–1617, 1638–1641 and 1899–1901), and torrid summers driven by low precipitation and exceedingly high temperatures (1527–1529, 1720–1722, 1813–1814, 1856–1857 and 1926–1930). More recently, Wan et al. (2018) analysed drought and flood data derived from documentary sources for the Baoji area (western Guanzhong region) in the period 1368–1911. Droughts occurred 191 times and there were 106 flood events. Relatively dry patterns were recognised between 1368 and 1644, followed by a phase of droughts and floods to 1804 and

comparatively wet patterns thereafter.

Zheng et al. (2006) used Chinese historical documents and instrumental measurements for the statistical reconstruction of a proxy precipitation index dataset for the period AD 501–2000 over Eastern China (*c*. 25–40°N, east of 105°E). For the whole of Eastern China they disclosed long-term dry periods in the 530s–570s, 640s–700s, 750s–800s, 840s–870s, 1000s–1230s, 1340s–1360s, 1430s–1570s, 1620s–1640s, 1920s–1930s, and from the 1980s. The most severe drought occurred in 1634–1644,

particularly on the North China Plain (*c*. 34–40°N) and in the Jiang-Huai area (*c*. 31–34°N), also extending to Jiang-Nan (*c*. 25–31°N). Other papers by Zheng et al. (2018) used the Yu-Xue-Fen-Cun historical archive, containing quantitative records of the depth of infiltration into the soil after each rainfall event, or the depth of each snowfall, combined with instrumental data for the reconstruction of seasonal precipitation at 17 sites in Northern China, for 1736–2000. They identified 29 extreme droughts, with higher drought frequencies during the 1770s–1780s, 1870s, 1900s–1920s, 1940s and 1980s–1990s. A higher probability of drought

occurrence was related to El Niño during the year in question or the previous year.

Ge et al. (2016) reviewed advances in high-resolution temperature and precipitation reconstructions over China for the past 2000 years. For Eastern China, they reported more frequent extreme droughts in the periods AD 301–400, 751–800, 1051–1150, 1501–1550 and 1601–1650. Extreme droughts and floods occurred together most frequently between 1551 and 1600. For the area between the north-eastern Qunghai-Tibet Plateau and the western margins of the Qinling Mountains, they mentioned several

multi-decadal severe droughts during the past millennium, in particular during the 1480s and 1710s. The arid and semi-arid zones of north-western China were relatively dry between 1000 and 1350.

Because of the long tradition of writing systematic official and local chronicles in the Chinese dynasties (since ~200 BC), a team of researchers led by Zhang De'er compiled all meteorological as well as weather- and climate-related records (harvest, famine, flood, drought, etc.) records from 7,930 historical documents into a compendium of Chinese meteorological records for the

past 3,000 years (Zhang, 2013; first edition 2004). The recently created REACHES database (see Sect. 2.15) may offer a significant new dataset for further climatological analyses (including drought), facilitating the use of computing technology for climate (drought) reconstructions in China.

Only a few other papers related to documentary-based droughts in Asia appertain to countries other than China. For example, Grotzfeld (1991) evaluated Arabic chronicles from the Near East between AD 800 and 1900 and presented (graphically

and with great time gaps) decadal frequencies of dry winter half-years for the regions of Iraq, Syria-Palestine and Egypt. A remarkable drought series for the period 1777–2008 is available for Korea, where Kim et al. (2011) used daily rainfall totals measured at Seoul with the traditional *chukwookee* Korean rain gauge between 1777 and 1907, supplemented by instrumental measurements from other stations. Using the Effective Drought Index (EDI), they identified 114 drought episodes from AD 1788 onwards, with the most extreme of them in 1899–1903.

**4.1.3 Africa**



Instrumental rainfall records for periods prior to the late 19th century are relatively sparse in Africa, with available data largely restricted to coastal locations (Nicholson et al., 2012a). The continent does, however, have rich written records, particularly from the late 18th century onwards, due to the activities of explorers, traders, settlers, missionaries and, later, colonial government officials, mainly European. Some sources provide extensive evidence for the spatio-temporal distribution and societal impacts of historical

droughts at inter-annual to multi-decadal scales. While, strictly speaking, an indicator of rainfall over the Nile catchment areas in Ethiopia (Blue Nile) and equatorial Africa (White Nile), Nilometer records from Cairo also provide a near-annually resolved drought chronology for north-eastern Africa dating back to the 7th century.

Drought chronologies derived from documentary sources exist at a variety of spatial scales from the regional to the continental. Arguably the most important studies for the continent as a whole are those of S. Nicholson. Her pioneering work since

the late 1970s has involved the compilation of historical accounts of drier and wetter conditions from across the continent, mainly from published late-18th- and 19th-century sources. Her basic methodology is described in Nicholson (1979, 1981a, 1996), with early sketches of the spatial extent of periods of unusual rainfall in Nicholson (1978, 1980). Other works detail historical fluctuations of lakes, mainly in eastern Africa (e.g. Nicholson, 1981b, 1998a, 1998b, 1999; Nicholson and Yin, 2001). The raw historical citations used in these studies have been combined into a database (see Nicholson, 2001b), and made accessible,

alongside early-19th-century rainfall data, via the World Data Service for Paleoclimatology in Boulder, Colorado (https://www.ncdc.noaa.gov/paleo/study/12201, last access: 27 August 2018; Nicholson et al., 2012a).

The most recent publications by Nicholson et al. (2012a, 2012b) and Nicholson (2014) have analysed combined documentary and gauge evidence to generate a seven-class index (from −3 to 3) time series of rainfall variability for 90 homogenous rainfall regions across mainland Africa for the period after 1800. These studies rely on the premise that information

pertaining to any location within a zone may be considered to represent wider conditions across that zone at that time. Statistical inference is also used to extend rainfall conditions from zones with available documentary or gauge data to adjacent zones that show high interzonal correlation (see Nicholson et al., 2012b). With the exception of northern Algeria and parts of eastern Africa, these studies reveal extensive aridity during the first three decades of the 19th century. Given the occurrence of widespread drought conditions during the 1790s (Nicholson, 1996, 2001a), this arid interval probably commenced around 1790. Most of the rainfall

zones also show a general tendency towards higher rainfall throughout the early- and mid-20th century, and later.

Other drought chronologies for Africa are either regional or sub-continental in scale. Despite some breaks, the longest available indirect record of drought for Africa is the Nile flood series (see Toussoun, 1925; Ghaleb, 1951; Popper, 1951; Hurst, 1952; Hassan, 1981, 2007b; Kondrashov et al., 2005). Analyses of calibrated Nile minimum flood levels reveal that discharge during the Medieval Climate Anomaly (MCA) was not constant. Instead, this period was characterised by episodes of considerably

low flood levels from 930 to 1070, and from 1180 to 1350 (Hassan, 2007b), reflecting drier conditions over the catchment. The start of the MCA was particularly marked by a dramatic increase in the frequency of extreme low floods (Hassan, 2011). Periods of low flood discharge are also identifiable in Nilometer records from 1470 to c.1500 and from 1725 to 1800 (Hassan, 2007b).

For West Africa, Norrgård (2015), analysed various British documents from locations along the Guinea coast of present-day Ghana, to establish drier and wetter periods from 1750 to 1800. These were compared against analyses of oral traditions and

Arabic chronicles from the Sahel (e.g. Cissoko, 1968; Curtin, 1975; Nicholson, 1978). In the Guinea coastal interior, drier phases occurred between the 1740s and mid-1750s, between the mid-1760s and early 1780s (with several famines and shortages of corn), and finally in the 1790s, again with famines and corn shortages. The last two dry periods also affected the Sahel area, while the coastal dry zone experienced severe drought in 1777–1787.

Regional rainfall chronologies are at their most concentrated for the southern African summer rainfall zone. Most of these

chronologies are based solely upon written sources and, in contrast to Nicholson, utilise gauge data only for validation and/or





calibration. They also use a five-point scale (–2 to +2) to classify rainfall, and have employed a relatively non-standard methodology for the classification of drier and wetter seasons (Nash, 2017). In the majority of studies, the average rainfall conditions for a specific season in a region are determined via qualitative analysis of the collective documentary evidence for that season. This contrasts with Nicholson's approach, which attributes a relative numerical score (–3 to 3) to individual quotations

according to how wet or dry conditions appear to have been; overall conditions during a season are then determined by averaging these scores. The discrepancies introduced by such different approaches are considered below.

      The earliest region-specific rainfall reconstruction for southern Africa is by Vogel (1989), who analysed published missionary diaries, monographs and British colonial *Blue Books* to develop an annual drought/flood chronology for the southern and eastern Cape. In the southern Cape, drier years and severe droughts were identified in 1825–1826, 1837–1839, 1846–1849,

1851, 1854, 1860, 1865, 1873–1875, 1880–1882, 1884–1885, 1892, and 1894–1900. An equivalent series for the eastern Cape reveals severe droughts in 1833–1834, 1837–1839, 1859, 1861–1862, 1865, 1877–1878 and 1895–1896.

      Nash and Endfield (2002a, 2002b) and Endfield and Nash (2002) applied Vogel's methodology to analysis of correspondence, reports and personal papers from British mission stations and other sites in order to reconstruct climatic variability and river flows in the Kalahari Desert of Botswana and northern South Africa from 1815 to 1900. They detected major dry periods

in 1820–1827, 1831–1835, 1844–1851, 1857–1865, 1877–1886 and 1894–1899, of which the droughts of 1844–1851, 1857–1865, 1884–1886 and 1894–1899 were the most widespread. A causal relationship to El Niño events was also suggested. This was further explored by Nash and Endfield (2008). Using maps of historical rainfall variability and time-sequences of documentary evidence, they established that the relationship between ENSO (El Niño–Southern Oscillation) and rainfall variability identified for the 20th century, whereby ENSO warm events are often preceded by wetter conditions and succeeded by drought, has held for much of the

last 170 years.

      Using analyses of missionary journals, letters, traveller's writings, and governmental reports, Kelso and Vogel (2007) identified major droughts in 1820–1821, 1825–1827, 1834–1836, 1855–1858, 1860–1862, 1865–1868, 1874–1875, 1880–1883, and 1893–1896 in semi-arid Namaqualand in north-western South Africa. They also suggested that some of these droughts were linked to ENSO episodes. A significant methodological advance in this study was the inclusion of "confidence ratings" for annual

classifications, based on the number and/or quality of historical sources; this approach has been adopted in subsequent reconstructions.

      In Lesotho, Nash and Grab (2010) used letters, journals and reports written by British and French missionaries and colonial authorities to classify rainfall levels from 1824–1900. They detected drought episodes in 1833–1834, 1841–1842, 1845–1847, 1848–1851, 1858–1863, 1865–1869, 1876–1880, 1882–1885 and 1895–1899, with the most severe droughts in 1850–1851

and 1862–1863. Seven of the reported multi-annual droughts coincided with periods of low ENSO values, and a tentative connection with Indian Ocean sea surface temperature (SST) variations was suggested. Significantly, this was the first study in the region to draw upon indigenous written sources, in the form of the Lesotho-language newspaper *Leselinyana la Lesotho*.

      Nash et al. (2016b) analysed 19th-century rainfall variability in the adjacent region of KwaZulu–Natal, South Africa, using English-, German- and Norwegian-language newspapers, colonial records and missionary materials. They identified eight

multi-year droughts between 1836 and 1900 in the rainy seasons of 1836–1838, 1861–1863, 1865–1866, 1868–1870, 1876–1879, 1883–1885, 1886–1890 and 1895–1900. Grab and Zumthurm (2018) used various documentary sources to construct a hydroclimatic history of central Namibia from 1845–1900. Using a similar approach to Nash and Endfield (2002a), they identified a slightly higher proportion of wetter years (42%) than dry (38%). In total, 19 dry rainfall seasons occurred; while the most severe drought, which included four consecutive "very dry" seasons, was recorded in 1865–1869.





Using historical documentary materials from British and African archives, Nash et al. (2018) reconstructed rainfall variability in Malawi for the latter 19th century. They identified widespread and severe droughts during the rainy seasons of 1861–1863, 1877–1879, 1885–1888, and 1892–1894. The El Niño event of 1877 was particularly strong, and was associated with drought from northern Malawi to the Eastern Cape, while drier conditions during a strong El Niño in 1855 extended from Malawi as far as the southern Kalahari. The authors compared their reconstruction with results for the equivalent rainfall zones in Nicholson et al. (2012b). The extreme droughts of the early 1860s, mid- to late 1870s, and mid- to late 1880s are visible in both reconstructions. However, discrepancies were identified in other decades, with the Nicholson et al. (2012b) series appearing to overestimate drier conditions, at least for parts of southern Africa.

Hannaford et al. (2015) were the first authors to use data from digitised ships' logbooks to reconstruct precipitation in Southern Africa, focussing their analysis upon four weather stations in the Eastern Cape and KwaZulu–Natal for the period 1796–1854. The reconstructions show a degree of correspondence with the regional drought series described above. For example, the mid-1820s to mid-1830s were the driest of the period at Mthatha (Eastern Cape) and Royal National Park (KwaZulu–Natal). At Cape Town, drier conditions were observed in the 1830s to the early 1840s. However, there are discrepancies between the authors' results and those for equivalent zones in the Nicholson et al. (2012b) synthesis. Nicholson et al. (2012b), for example, indicate protracted drought for 1800–1811 in "South Central Africa", while Hannaford et al. (2015) and a tree-ring series for Zimbabwe (Therrell et al., 2006) identify above-mean rainfall alternating with drought during this period.

Using a more qualitative approach, Hannaford and Nash (2016) have analysed fragmentary Portuguese accounts of climate-related phenomena in south-eastern Africa between ~1500 and 1830. They identified five clusters of widespread, severe and protracted drought events. The period 1506–1518 saw drought and food shortages along the Mozambique coast, and in 1730–1768 drought, locust plagues and famine along the Zambezi valley. Southeast Africa-wide droughts, with associated food scarcity and famine, were identified in 1560–1590, 1795–1805 and 1824–1830, independently corroborating many of the drought chronologies noted above.

### 4.1.4 The Americas and Australia

For North America, Mendoza et al. (2005a) addressed droughts in central Mexico on the basis of a catalogue of agricultural disasters derived from documentary evidence for the period 1450–1900. From a total of 388 documented droughts they compiled a series of 70 droughts. Periods of severe drought centred around the years 1483, 1533, 1571, 1601, 1650, 1691, 1730, 1783, 1818 and 1860. In a subsequent paper, Mendoza et al. (2006), analysing largely those droughts affecting agriculture in south-eastern Mexico, reported the highest drought frequency around 1650, 1782 and 1884, while no drought reports are known from *c.* 1540, 1630–1640, 1672–1714 and 1740–1760. Clearly more droughts occurred between 1760 and 1899 than between 1550 and 1760. The droughts, usually lasting one or two years, appeared in various cycles. Mendoza et al. (2007) used documentary evidence to study the frequency and duration of droughts in the Mexican Maya lands on the Yucatan Peninsula for the period 1501–1900. Severe droughts were detected only for 1648–1661, 1725–1727, 1765–1773 and particularly between 1800 and 1850. They usually lasted for one year at a quasi-decadal frequency.

In the U.S.A., Dupigny-Giroux (2009) used daily journal entries covering the 1680–1900 period in the New England states of Vermont and New Hampshire to investigate droughts (among other climatic matters). She developed a qualitative drought index with categories from S1 ("flash" droughts of 1–2 months duration) to S5 (multi-year, severe droughts). The region was especially drought-prone in the late 1700s, with a shift to a wetter climate around 1850. Multi-year, severe droughts (S4, S5) were particularly evident in 1762, 1794, 1849 and 1864.





Guevara-Murua et al. (2018) derived information from Catholic rogation ceremonies, together with flooding events and crop shortages recorded in the minutes of meetings of the city and municipal councils in Antigua Guatemala and Guatemala City on the Pacific coast of Central America in the 1640–1945 period. Classifying patterns of rainy seasons from May to October in a five-point scale, from very wet to very dry, they detected 34 years of drier conditions and 21 years of very dry conditions. Drier

periods occurred between the 1640s and the 1740s (related to the southward displacement of the Intertropical Convergence Zone), in the 1820s and the 1840s. Berland et al. (2013) used over 13,250 items of documentation (missionary, plantation and government papers, as well as contemporary scholarly publications) from Antigua (Lesser Antilles) to create a five-degree classification (very wet, wet, normal, dry, very dry) of precipitation patterns, subsequently applied in the reconstruction of rainfall variability in the period 1770–1890. Significant dry phases were identified in the rain-years 1775–1780, 1788–1791, 1820–1822, 1834–1837, 1844–

1845, 1859–1860, 1862–1864, 1870–1874 and 1881–1882.

In South America, Prieto et al. (2000) used documentary data from Bolivia and Argentina from the 17th and 18th centuries to provide new information relevant to the great droughts in the Andes between 1780 and 1810, identifying that heavy rains and extreme floods occurred frequently in the Río de la Plata basin. Prieto and Rojas (2015) reconstructed a series of droughts and high water flows on the Bermejo River (Argentina) during the 17th–20th centuries on the basis of a content analysis of

documentary evidence, calibrated against instrumental data. They identified a significant decrease in extreme droughts that began in 1890. Longer dry periods were identified between the 1680s and 1710, while episodic droughts occurred in the 1720s, 1760s, 1780s, 1790s, 1870s, 1890s and 2000s. These results correlated well with fluctuations of the River Mendoza. Domínguez-Castro et al. (2018) studied wet and dry extremes in Quito (Ecuador) using rogation ceremonies in the Chapter Acts of Quito (1600–1822), and instrumental rainfall series (1891–2015) for which SPI had been calculated. Rogations proved a sound proxy for droughts and

extreme wet conditions during January–March and wet conditions during June–August. The most important drought periods occurred in 1692–1701 (the severest drought), 1718–1723, 1976–1980, 1990–1993 and 2001–2006. The frequency and duration of droughts appeared to intensify, particularly from the mid-20th century.

In Australia, Nicholls (1988) used early documentary records (mainly letters) from the British colony of New South Wales to identify droughts in the 1788–1841 period. Only three droughts (1796–1797, 1798–1799 and 1810–1811) were not associated

with ENSO events, compared with a further eight droughts detected (1790–1791, 1803–1804, 1813–1815, 1817–1818, 1819–1820, 1823–1824, 1826–1829 and 1837–1839). Later, Fenby and Gergis (2013) used 12 secondary documentary sources, starting with the first European settlement in Australia in 1788, and continuing through widespread meteorological observations from 1860 onwards, to create 12 documentary-based rainfall chronologies for south-eastern Australia. They identified 27 drought years in the whole period. The longest and most widespread event, influencing all subregions, occurred in 1837–1841, but periods of

considerable drought were also detected in some parts of south-eastern Australia between (1825–)1826 and 1830, as well as in 1809–1812 and 1813–1816. The results of this paper were used by Gergis and Ashcroft (2012) to create an extended rainfall index for eastern New South Wales for 1788–2008, in which they identified 81 dry years (Fig. 7). Ashcroft et al. (2014) analysed early meteorological observations from 39 archival sources in south-eastern Australia for the 1788–1859 period. Establishing regional means, they identified prolonged dry conditions in various parts of the region during 1837–1843 and 1845–1852.

**4.2 Individual and major drought events**

Pankhurst (1966) reported 1888, a year of major El Niño, as excessively dry and hot in Ethiopia, where lack of rain gave rise to severe harvest failure resulting in famine the following year, exacerbated by a major epidemic of cattle plague (rinderpest) and an outbreak of locusts and caterpillars. The famine and subsequent epidemics resulted in the deaths of around a third of the entire population of the country (see also Pankurst, 1985). Nicholls (1997) also reported considerable drought damage in Brazil and





Australia in this El Niño year, but of lesser extent than that occurring in Ethiopia. Wolde-Georgis (1997) added a list of El Niño events, droughts and famine in Ethiopia, covering the years from the 1540s onwards.

Many of the major drought episodes to affect Africa have already been described in Sect. 4.1.3. For southern Africa, Neukom et al. (2014) combined available annually-resolved documentary, early-instrumental, tree-ring and coral ($\delta^{18}$O, Sr/Ca)

series into a multiproxy reconstruction of summer and winter precipitation for the past 200 years. The summer rainfall zone reconstruction, updated by Nash et al. (2016b) to include new rainfall series, identifies coherent dry intervals at around 1827 and 1862, the latter the driest year in the 19th century. The 19th century was significantly wetter than the 20th, with this drying trend continuing into the early 21st century. The driest years of the 19th (20th) century in the winter rainfall zone include 1825, 1827 and 1865 (1935, 1960 and 1973), with rainfall becoming more variable during the 20th century (Neukom et al., 2014).

For the years following the prolonged Eldgjá volcanic eruption in Iceland, starting around AD 934 and continuing for 3–8 years, Fei (2006) reported for China a number of anomalous weather conditions associated with extremes that led to hardships. This was followed by an extraordinary drought and locust invasion in AD 942–943, with widespread famine through the Yellow River Basin and the Yangtze River Basin, resulting in the deaths of several hundred thousand people. This drought was at least partly responsible for the collapse of the Late Jin Dynasty.

Dodds et al. (2009) reconstructed spatial patterns of drought and hurricane tracks in the U.S. for 1860. Using instrumental data together with documentary accounts from diaries, newspapers and ships' logbooks, drought conditions were identified across much of the central and southern Plains, parts of the Midwest and in the south-western U.S.A. The central plains of Kansas, Oklahoma and Missouri made up the core of a region where dry or very dry conditions prevailed from April through October and where impacts on crops were at their most severe. The rainfall associated with three land-falling hurricanes in the south-eastern

U.S.A. came too late for crops to recover.

Kiss (2009) provided an overview of selected Hungarian scientific literature related to drought. She drew attention to, for example, such outstanding droughts as those of 1717–1718 and 1728–1729 (notably blamed for the infamous witch-hunt of Szeged – see Sect. 4.4.3), and the early 1790s drought that resulted in a crisis of large-scale animal husbandry on the Great Hungarian Plain. She also described the extremely severe drought of 1863 that catalysed, for example, changes in agricultural practices and

the establishment of the Hungarian Meteorological Service.

Hao et al. (2010) performed a detailed analysis of the most severe and extreme drought for the past 300 years in Northern China, an event that started in spring 1876 and continued until spring 1878. Harvest failure during these three years increased rice prices to five or ten times over those for a normal year; as a result, more than 20 million people in the five provinces of Northern China died or emigrated. According to the authors, this drought prevailed worldwide with a possible link to abnormally high SST

in the equatorial central and eastern Pacific Ocean, strong El Niño and positive Antarctic Atmospheric Oscillation (AAO) anomalies. The spatial extension and dynamic evolution of the large-scale 1876–1878 drought and its major consequences are examined by Zhang and Liang (2010). This persistent drought disaster hit 13 provinces. In Shaanxi, Henan and Shanxi provinces over 340 days passed without soaking rain, suggesting that it was a more severe event than even the greatest drought of the 20th century (between 1928 and 1930).

Wetter et al. (2014) used more than 300 first-hand documentary sources of weather reports originating from an area of 2–3 million km² in Europe to describe an unprecedented 11-month "megadrought" in 1540. With the exception of winter, the estimated number of precipitation days and precipitation totals of the individual seasons for Central and Western Europe were significantly lower than the 100-year minima derived from instrumental measurements. Information about the drought was supported by independent documentary evidence reporting extremely low river flows and forest fires, settlement fires and wildfires.



Kiss and Nikolić (2015) analysed the great droughts of 1362 in Dalmatia (Croatia) and in 1474, 1479, 1494 and 1507 in Hungary, characterised by lack of precipitation, water shortages, extremely low water levels in major rivers, bad harvests and severe food shortages, often accompanied by, or followed by, locust outbreaks. In a subsequent paper, Kiss (2017) extended this analysis to other episodes of drought and low water levels in medieval Hungary for 1361, 1439, 1443–1444, 1455, 1473, 1480, 1482, 1502–1503 and 1506, comparing them with a hydroclimate reconstruction of the OWDA by Cook et al. (2015). Kiss (2017) pointed out a close agreement of OWDA evidence with documentary-based Hungarian drought records, only exceptionally detecting differences in spatial extension and drought intensity (e.g. in 1455 and 1507). Kiss (2018) analysed the spatial distribution, physical characteristics and documented socio-economic impacts, including harvest failures, livestock problems, material damage and market reactions, associated with the (1506–)1507 drought that occurred in the Carpathian Basin. Further, she discussed the complex relationship between drought, its multi-annual consequences, the increase of plague and problems along the southern military defence line, together with the detectable individual and institutional responses (e.g. tax exemption, charity, public works).

Roggenkamp and Herget (2015) reported a drought in the year AD 69 based on the fifth book of Tacitus' *History*, in which he mentioned a battle between Roman troops and the Germanic Bataver tribe that took place in the channel of the River Rhine (near today's Krefeld in Germany). A Roman transport ship ran aground in the shallows for lack of water and was attacked by the tribes. Using Tacitus' descriptions and archaeological findings, Roggenkamp and Herget (2015) reconstructed the Lower Rhine discharge at the time as less than 300 m$^3$ s$^{-1}$, much lower than the lowest measured in the particularly dry year of 1947 (530 m$^3$ s$^{-1}$). They hypothesized that droughts, because of the trouble they brought to transport, led to more economic damage and social conflicts in the Roman Empire than flood events.

Using Czech documentary evidence, Munzar (2004) examined five extreme droughts in the Czech Lands and neighbouring countries that occurred in 1540, 1590, 1616, 1790 and 1842. The extreme drought of 1616 in the Czech Lands and some surrounding countries was again reported by Munzar and Ondráček (2016). However, documentary data may permit the more complex characterisation of extreme droughts during the instrumental periods, as illustrated by the example of the catastrophic 1947 drought in Central Europe (Brázdil et al., 2016b).

Grove and Adamson (2018) associated very prolonged, exceptional droughts in southern Asia, and also in Australia, America and Africa, with the "Great" El Niño of 1790–1794. Documentary evidence suggests that this may have been one of the most severe drought periods on written record.

### 4.3 Droughts, external forcing and large-scale climate drivers

As the effects of external forcing and large-scale climate drivers on the occurrence and severity of droughts can be geographically very variable, this section presents them separately for broader regions.

### 4.3.1 Europe

Relatively few studies have investigated above effects for drought series in Europe (e.g. Pongrácz et al., 2003). Brázdil et al. (2015) studied effects of external forcing and large-scale climate drivers in six spring–summer drought indices series (SPI-1, SPI-12, SPEI-1, SPEI-12, Z-index and PDSI) for the Czech Lands for 1805–2012. They identified the importance of the North Atlantic Oscillation (NAO) phase and the aggregate effect of anthropogenic forcing (increasing CO$_2$ concentrations). These effects were strongly dependent on the type of drought index and season. Solar irradiation and the Southern Oscillation Index (SOI) made only minor contributions to central European drought variability, while the effects of volcanic activity and the Atlantic Multidecadal Oscillation (AMO) were even weaker and statistically insignificant. Mikšovský et al. (2018) studied forcings in reconstructed 500-



year SPI, SPEI and PDSI series from the Czech Lands (Brázdil et al. 2016a) and confirmed the importance of anthropogenic forcing, while AMO and Pacific Decadal Oscillation (PDO) indices proved important internal factors, accounting for significant contributions to inter-decadal drought variability. Colder and wetter episodes were found to coincide with increased volcanic activity, while no clear signature of solar activity emerged. However, working with drought series for central-southern Italy

spanning the period 1581–2007, Diodato and Bellocchi (2011) disclosed distinct 11-year and 22-year cycles that could reflect single and double sunspot cycles. They argued that low sunspot activity (e.g. the Maunder Minimum) could have more effect on drought than local forcing agents. Vicente-Serrano and Cuadrat (2007) noted that positive values of the NAO had a strong effect on a December–August drought index compiled from rogation ceremonies in the semi-arid middle Ebro valley region of north-east Spain between 1600 and 2000.

**4.3.2 Asia**

Tong et al. (2006) studied the relationship between floods, droughts and ENSO activity over the past half-millennium in China. While droughts correlated significantly with La Niña and floods with El Niño events in the lower and middle catchment of the Yangtze River, these relationships were reversed in the Upper Yangtze catchment. Cycles of drought periods were shown to be longer than those of ENSO events. Similarly, Jiang et al. (2006) analysed the teleconnections between flood/drought index series

for the Yangtze River valley in 1470–2003 and ENSO. In the lower and middle Yangtze catchment, droughts correlated with La Niña phases, but in the upper part with El Niño phases. ENSO changes and flood/drought variation were significantly correlated at *c*. 5-year and *c*. 10–12-year cycles. Ji-bin et al. (2006) analysed dry-wet development in Guangdong Province over the last 500 years and found correlations between droughts and El Niño events in its western and northern parts.

For Eastern China, Shen et al. (2007) identified three exceptional drought events within the past 500 years: 1586–1589,

1638–1641, and 1965–1966. The most recent event arose out of a weakening of the summer monsoon and an anomalous westward and northward displacement of the western Pacific subtropical high. Moreover, these three exceptional drought events could have been triggered by large volcanic eruptions and amplified by the combination of volcanic eruptions and El Niño events. In a subsequent paper, Shen et al. (2008) used a documentary-based dryness/wetness index to analyse anomalous precipitation events for Northern China and the middle-lower Yangtze River valley, addressing the past five centuries. A high frequency of occurrence

of such events coincided with periods of high solar forcing, active volcanic eruptions, and considerable anthropogenic forcing (during the 20th century). Coherent droughts occurred frequently in the 17th and 20th centuries; they were also significantly associated with explosive low-latitude volcanic eruptions. Similarly, Ge et al. (2016) mentioned the possible triggering effect of large volcanic eruptions, amplified by the combination of volcanic activity and El Niño events, in the occurrence of exceptional droughts in Eastern China in 1586–1589, 1638–1641, 1876–1878 and 1965–1966. Wan et al. (2018) found correspondence with

sunspot cycles for three drought/flood cycles in the Baoji area in the 1368–1911 period.

Yang et al. (2014) analysed the variability of droughts at the northern fringe of the Asia summer monsoon region (NASM), including China, over the last two millennia. Using tree-rings, instrumental and documentary (dryness/wetness) data, they identified two periods of drought in AD 1625–1644 and 1975–1999, with a wet west and a dry east in the NASM. Thus, when drought was observed in the Karakorum, the Western Tien-Shan, the Pamir, in Mongolia, most of eastern Asia, the eastern Himalayas and south-eastern Asia, wet conditions prevailed on the Indian subcontinent. The authors argue for a combined effect of

a weakened Asian summer monsoon and an associated southward shift of the Pacific Intertropical Convergence Zone in 1625–1644, while droughts in 1975–1999 were associated with warm temperature anomalies in the tropical Pacific.

**4.3.3 Africa**





A number of studies from Africa have noted the influence of global climate forcings upon historical droughts (see Sect. 4.1.3). For example, several papers have demonstrated a significant association between rainfall in Nile tributary catchments, the Indian monsoon, and ENSO (e.g. Walker, 1910; Quinn, 1992; Zickfeld et al., 2005). Lindesay and Vogel (1990) identified a close association between droughts and El Niño events by analysing series from the eastern Cape in South Africa produced by Vogel

(1989). Comparison of variations in Nile flood discharge with NAO proxies also reveals a strong coherence between discharge fluctuations and variations in North Atlantic SSTs (Jansen and Koç, 2000).

Multiproxy reconstructions of southern African rainfall by Neukom et al. (2014) and Nash et al. (2016a) have been correlated against a number of large-scale climate indices, including the SOI, Southern Annular Mode (SAM) and the second principal component of the Indian Ocean SSTs. The SOI and Indian Ocean SSTs appear to have a stronger influence on summer

rainfall zone precipitation than SAM. However, while the correlation coefficients for these indices are relatively consistent over time, there are breakdowns in the SOI-summer rainfall zone precipitation relationship from *c*. 1830–1875 and *c*. 1930–1960. The timing of these breakdowns tallies with analyses of instrumental data and palaeo-data from elsewhere around the Indian Ocean rim, suggesting a basin-wide weakening of ENSO teleconnections during the mid-19th and mid-20th centuries (Nash et al., 2016a). Relationships between winter rainfall zone precipitation and the SOI and SAM over time are weak (Neukom et al., 2014).

**4.3.4 The Americas and Australia**

A number of papers address forcings in drought series in Mexico. For example, Mendoza et al. (2005a) found a significant link between droughts and El Niño events of all intensities for central Mexico. In a further paper, Mendoza et al. (2005b) used a catalogue of documentary-based droughts in central and south-eastern Mexico, derived from agricultural disasters, to identify the possible effects of various solar activity phenomena during the 1400–1900 period. Detecting a range of cycles in both regions, they

concluded that a statistically significant connection exists between drought occurrence and solar activity (solar irradiance in particular). Turning to south-eastern Mexico, Mendoza et al. (2006) found a relevant relationship between El Niño events and droughts and – depending on the time period – a somewhat stronger relationship with changes in SST and SOI, as well as solar activity. For the Yucatan Peninsula, Mendoza et al. (2007) found a coincidence of historical droughts with the cold phase of AMO, although the influence of SOI was less clear.

Prieto et al. (2000) established, in a study of documentary-based climate variability during the 17th–18th centuries, a connection between ENSO and extraordinary droughts in Bolivia and north-western Argentina, the abundant rainfall in Mendoza and floods in north-eastern Argentina. Gergis and Aschcroft (2012) used an extended rainfall index for eastern New South Wales (Australia) for 1788–2008, developed from documentary, early instrumental and modern observations, for comparison with ENSO series. Although connections between ENSO and rainfall variability exist there, only a weak relationship was identified in the

coastal part of the area studied.

**4.4 Droughts, socio-economic impacts and human responses**

Documentary sources, and the papers that analyse them describe a broad variety of drought impacts on human society and associated responses, which are considered below, thematically arranged.

**4.4.1 Droughts, harvest failure, famine, disease, invasion and migration**

Bad harvests or even failure of agricultural crops, followed by increases in food prices, shortages, hunger, epidemic and famine, appear among the most extensive impacts of droughts, identified in many studies employing documentary data. For example,



Miller (1982), based on *c*. 170 references concerning drought and disease extracted from documentary sources, suggests that these factors had a significant impact on farmers and pastoral communities in West-Central Africa between 1550 and 1830. In general, only the longer and more severe droughts, of wider spatial extent, were recorded since they led to major socio-economic problems, including warfare. Such major droughts occurred in the late 16th and late 18th centuries. The Angolan slave trade appears to have

been a by-product of warfare associated with the former of these droughts, while the later drought period coincided with the peak of slave export from the region. Severe droughts were similarly blamed for playing a significant (if indirect) role in the burgeoning slave trade in, for example, the southern Red Sea region in the late 19th–early 20th century, and in colonies where agriculture was largely based on slave labour (e.g. Réunion, Mauritius and the Greater Caribbean in the 18th century; from Sudan to Egypt and the Ottoman Empire in the late 18th–19th century; see La Rue, 2010; Peabody, 2010; Serels, 2010; Johnson, 2012).

An anomalous period became evident in the 1780s and 1790s in Egypt after the great Laki (Lakagígar) eruption of 1783, characterised by low Nile floods, drought, harvest failure, famine and disease (Mikhail, 2015). These problems were compounded by consequent economic and political chaos. In parallel, severe droughts, arising out of strong La Niña activity, resulted in famine and the spread of smallpox in 1780–1782 in central North America, with the disease outbreak expanding to epidemic proportions (Hodge, 2012). Furthermore, the same author suggests that the unusually swift spread of smallpox among the transhumant tribes of

the northern Great Plains was due to the impact of drought and climate fluctuations upon grassland productivity and associated bison migration patterns.

     Tarhule and Woo (1997) looked into the relationship between measured rainfall, documented droughts and famines (using colonial administrative and ethnographic collection reports) in the sub-Saharan region of Nigeria for the 1905–1996 period. Their results suggest, with 90% certainty, that drought-induced famines occurred during periods of cumulative rainfall deficit. The study

also contains an appendix with extracts from historical droughts and famines from 1600 onwards, published in earlier works.

     A review paper by Fan (2015) provides an inside view of Chinese climate history research, including studies on historical droughts, with special emphasis on socio-economic consequences. It draws particular attention to the relationship between the climate (including droughts) that shaped and influenced agriculture and the wider economy, as well as riots, wars and nomad conquests in China, followed (or preceded) by dynastic decline. For example, Ye et al. (2012) considered more frequent climatic

disasters – particularly floods and droughts in 1851–1859, and droughts in 1875–1877 and 1927–1929 – as contributing factors to seven peaks of migration and reclamation in north-eastern China. According to Zheng et al. (2014), more bad harvests in the mid- and late 16th century, caused by cooling, aridification and desertification and an increase in the incidence of climatic extremes, had detrimental effects on the military farm system, raised army expenses and led to fiscal crisis. It has been proposed that severe droughts in 1627–1643 were largely responsible for famine and a subsequent peasant uprising. Su et al. (2014) reported poor grain

harvests in China between 206 BC and AD 960 arising out of cold and dry climatic conditions. Based on a combined tree-ring and historical documentary database for ancient China, covering 2000 years, Tian et al. (2017) argued that the likelihood of epidemics rose during cold and dry periods, due to higher frequencies of both locust invasion and famine. According to Li et al. (2017), frequent climatic extremes (such as floods and droughts) during the period 1760–1820, together with the pressure of population increase, resulted in the migration of people from Gansu and Shaanxi to Xinjiang. Liu et al. (2018) investigated what lay behind

the collapse of the late Ming Dynasty taking into account possible factors in the last 35 years of its existence (i.e. 1610–1644). Severe droughts, locust invasions and a cooling environment were identified as the main natural culprits, while the leading human elements included rebellions, inter-ethnic conflicts and a financial crisis. The authors conclude that natural factors may have played a somewhat more important role than human factors.

     An extreme drought, among other climatic extremes, played a significant role in the development of the Great Famine in

Iran in 1870–1872. No rain at all fell in winter 1869/1870, some came later, although only in the western and southern provinces,



but no rainfall occurred for two years in many areas. This resulted in harvest failures, even in irrigated lands, due to a lack of surface and subsurface water (Okazaki, 1986; Melville, 1988; Seyf, 2010).

According to Kiss and Nikolić (2015), severe, long-term droughts with prevailing low and very low water levels on major rivers in medieval Hungary and Slavonia (i.e. the Carpathian Basin) were also responsible for the weakening of the countries'

military defences, resulting in vulnerability to Ottoman-Turkish attacks. Noone et al. (2017) cited droughts in Ireland over the previous 250 years as the cause of agricultural hardship, water resource crises and failures, and subsequent major famine during the 18th and 19th centuries. According to Domínguez-Castro et al. (2018), the most severe drought of the past 400 years in Ecuador, in 1692–1701, led to a great famine, devastating Quito and affecting most of the central Andes. Grab and Zumthurm (2018) reported a semi-arid and water-deprived environment for central Namibia between 1845 and 1900 that resulted in severe social hardship,

mainly due to livestock mortality (e.g. lack of grazing, dehydration), crop failures and a lack of water for human consumption, leading in some cases to malnourishment and famine.

Even in instrumental period, documentary evidence may also prove valuable in the analysis of severe droughts and their impacts. For example, Liang et al. (2006) reported the devastating effect of a hard and sustained drought during the 1920s and early 1930s in northern China, impacting upon on agricultural productivity, hydrological resources, society, and natural vegetation;

they combined tree-ring, documentary and instrumental evidence. MacManus (2008) examined the causes and processes of a mass population exodus that occurred in south-western and west-central Saskatchewan (Canada) between 1917 and 1927. Drought and associated harvest failure were the main causes, with consequent bankruptcies and loss of property (i.e. land) to foreclosure. This crisis and abandonment process occurred before the great agricultural and economic depression of the 1930s, a direct continuation of the problems rooted in the first decades of the 20th century. According to Marchildon (2016), the great drought, combined with

the economic crash of the same period, was responsible for the greatest ecological and human disaster of the 20th century on the Canadian Prairies, particularly in its central part (the Pallisade Triangle), in the 1930s. Basing his work on documentary evidence, the author examined and summarised the administrative response, management strategies, and policy interventions of the governmental, federal, provincial and local administrations during this difficult period, in order to provide lessons for the future. Similarly, Brázdil et al. (2016b) used documentary evidence to characterise the catastrophic drought of 1947 and its broad

economic and political impacts and consequences in central Europe, with particular attention to the Czech Lands.

However, care is required before attributing agricultural and associated societal crises to episodes of drier climate alone. In south-eastern Africa, for example, climatic narratives have long been put forward to underpin the emergence and later dominance of the politically centralised Zulu kingdom during the late 18th and early 19th centuries. Hall (1976) was the first to hypothesise that the rise of the Zulu kingdom was an end-product of a period of warmer and wetter climate and population

expansion during the mid- to late-18th century (during which drought-vulnerable maize was introduced as a staple crop), followed by droughts and associated conflict around the turn of the 19th century. Contemporary debates (summarised by Hannaford and Nash, 2016) instead suggest that changes in climate, agro-pastoral livelihoods and ritual power, together with responses to new trade opportunities, transformed societal contexts in the region from the late 18th century onwards. Perhaps the key message here is that droughts, even if extremely severe and protracted, may not be the sole drivers of socio-political crisis and change: multiple

interlinking factors are much more worthy of consideration.

**4.4.2 Droughts, ancient civilisations and colonial regions**

Particular attention has been devoted to the effects of drought upon the development (and decline) of ancient civilisations and once-colonial regions. For example, Schneider and Adalı (2014) attributed the decline of the Neo-Assyrian Empire – aside from military conflicts – to population increase (making society more vulnerable to the negative impacts of drought) on the one hand,



and to an intensive period of drought in the mid-7th century BC with harvest problems on the other. As with the example from Zululand noted above, both factors might have resulted in the economic and political weakening of the empire, which could then not withstand the onslaught of the Babylonian-Median coalition in 612 BC. Low floods on the Nile, as indicated by palaeoclimatic and documentary evidence, were considered prime reasons for the great Egyptian famines, related riots, disorder and the fall of

dynasties witnessed around 2200 BC and AD 963 (Hassan, 2007a). Similarly flood failures induced by droughts after major volcanic eruptions have been blamed for revolts and interstate conflicts in Ptolemaic Egypt (Manning et al., 2017).

In Southern Mexico, Endfield et al. (2004) examined the consequences and societal responses of the colonial Oaxaca in relation to weather extremes, covering the period from the mid/late-16th to the early 19th centuries, based on primary and secondary documentary sources. Real crises tended to develop when harvest failure, often induced by drought (or other weather-

related hazards) affected larger areas and communities, and especially when repeated and/or multi-annual drought (and other) problems occurred. The authors noted that conflict over the possession and use of water supplies increased during periods of drought; and that, aside from the negative effects of weather extremes, the number of harvest failures may have been increased by high-risk cultivation practices. Endfield et al. 2004) also investigated community and administrative responses to drought and other weather extremes in Mexico. Using archival documentation, Endfield and Tejedo (2006) discussed the consequences of, and

societal responses to, prolonged droughts in late colonial Chihuahua during the 18th–19th centuries. Droughts were clearly responsible for harvest failure, famine (humans, domestic animals) and problems in mining activities in the 1720s, 1739–1741, 1748–1752, 1755, 1758, 1760–1766, 1770–1773, 1785–1786, 1804–1806, 1809 and 1812–1814. Furthermore, periods of social unrest often coincided with food shortages, being therefore indirectly related to droughts. Burns et al. (2014) used documentary data and tree-rings to identify the most important causes of the emergence and spread of typhus in Mexico between 1655 and 1918,

identifying great historical drought events as key factors. The environmental crises consequent upon severe drought led to famines that forced many people from the countryside into towns, where their inadequate living conditions and huge numbers greatly increased the risk of epidemic disease. White (2014) investigated the relationship between climate-related natural hazards and the conflicts between colonists and indigenous Pueblo communities in North America. Unexpected (and often unprecedented) cold or drought conditions and famine acted as catalysts in conflicts between colonists and indigenous people and could lead to the

demoralization of soldiers and settlers; these had negative effects on more intensive migration or investment, but indirectly encouraged missionary activities. The Pueblo Revolt was also fuelled by drought and famine.

Gergis et al. (2010) examined the impact of weather and related natural hazards on the development of colonial settlements and agriculture in Australia, with special consideration of the impacts of the 1791–1793 droughts. The authors investigated the way in which water scarcity influenced society and socio-economic processes in Australia in the late 18th century.

Fenby et al. (2014) concentrated on the droughts and floods that occurred in the first decades of the British colonisation of Australia between 1788 and 1815, together with their impacts on socio-economic conditions, on the developing settlements, on networks and on agriculture.

### 4.4.3 Perceptions of drought and spiritual approaches to it

While the majority of social and economic systems have evolved over time to accommodate some deviations from "normal"

weather conditions, this rarely holds true for extreme events such as severe and/or protracted droughts (Endfield and Veale, 2018). For this reason, such events may have the greatest and most immediate impact of all climate changes. For many cultures, drought memories are embedded in folklore and form the basis of ritual and religious practices. Examples of these practices, preserved in historical documentation, are summarised below.



Many agriculturalist African cultures and societies have long-standing traditions relating to drought and other types of climate variability (Parrinder, 1974; Ombati, 2017). One of the most important is rainmaking, whereby prayers and other rituals are offered or performed to various creator beings to encourage rainfall at time of moisture deficit or delayed seasonal rains. Whitelaw (2017), drawing upon oral histories and archaeological evidence, documents rainmaking practices across Nguni-speaking areas of southern Africa, where rainmaking remains a specialised, inherited profession passed down through blood and marriage. In these areas, ensuring rainfall was the traditional responsibility of the chieftaincy, with professional rainmakers called upon in times of drought to interact with extra-social forces that could deliver both "creative energy" in the form of rain, and destructive conditions such as drought, lightning and hailstorms. Whitelaw (2017) emphasises that rainmaking was a process that was integrated into annual agricultural cycles, arguably because the chief's fortune depended upon the smooth operation of the cycles and associated rituals. However, as Klein et al. (2018) note, power over the elements was not restricted to rain *making*, citing evidence of rain *control* rituals among groups in south-eastern Africa as a means of preventing rain from falling over enemy lands.

Human societies have adopted similar responses to droughts in other parts of the world, with the aim of diminishing their negative impacts. Rainmaking experiments among settlers in North Otago (New Zealand) during great droughts were described by Beattie (2004), who divided scientific and religious concepts and practices. Residents applied prayer and experiments in parallel, and – despite all theories – there was no sharp division between using a religious and practical/scientific approach, suggesting a higher level of religious tolerance in New Zealand than, for example, in England or Australia. In a later paper, Beattie (2014) provided case studies of the greatest drought events, documented in 1889–1891, 1906–1907 and 1909–1911, that led to bad harvests, cattle mortality or early slaughter and (in somewhat similar fashion to the South African case above) forced farmers to leave agriculture and take up other professions. The paper also examined how climate was included in the local religious and scientific debates of settlers and their causative interpretations.

According to court records of the infamous witchcraft trials in medieval Europe (e.g. Behringer, 1999; Pfister, 2007), witches were held responsible for changing the weather, including the summoning of drought phenomena. Zgutam (1977), who analysed Russian witchcraft in the 17th century, notes that inducing drought was among the most prevalent of accusations. As reported more recently by Levack (2016), men and women accused of bringing about drought in late medieval Russia were executed in such great numbers that reports appeared in annals and chronicles. Worobec (1995) discussed attitudes to the relationship between drought and witchcraft among Russian and Ukrainian peasants in the 19th and early 20th centuries. Again, one of the major accusations, especially during difficult periods, was that a witch had brought about the drought (Fig. 8). Times of increased social tension and decreased resources were especially hard for "witches", when community support for accusations was also acutely heightened. It was often expected that the accused individuals could, or would, reverse the spell. Witches were punished for allegedly creating dry weather in Upper Hungary and particularly in Poland. Brázdil et al. (2008b), analysing spring–autumn drought in Slovakia (former Upper Hungary), pointed out the persecution of witches in several places not only there, but also in other parts of Hungary. Witches were similarly blamed and punished for the 1726 drought in the Spiš area (Slovakia) and three years later a massive trial of weather-changing witches ended in Szeged (Hungary). The persecution of witches for causing great droughts and "taking away the dew" appeared in the biggest witchcraft trial of all, in early modern Hungary, which took place between 1718 and 1728 (Petrovics, 2005). According to Pócs (2005), droughts in Hungary appeared as the second most important weather disaster created by witches, second only to hailstorms. Such drought-related witch-hunt cases were mentioned, for example, in 1711, 1718, 1728, 1730 and 1758.

**4.4.4 Droughts, social resilience, public policy, administrative responses and their development**





Societies have always had to adapt to droughts, developing mechanisms or modes of behaviour to avoid the worst of their negative effects. In Europe, Taylor et al. (2009) discussed the socio-economic management of the seven greatest periods of drought and water scarcity during the 1893–2006 period in England and Wales. They argued that these events were not merely natural events, but were also shaped by the framework of institutional water management. The central priorities for water distribution were also

directly linked to the levels and severity of drought (as well as vulnerability) in different localities. In general, private systems reacted more flexibly than the public supply to the problems resulting from expanding water use. Grau-Satorras et al. (2016) analysed community responses, basing their work largely on the minutes of village councils, for Terrassa (Barcelona, Spain) during the 1605–1710 period. They concluded that decisions varied over time, and the symbolic/ritual, institutional and infrastructural (e.g. changes in water-related infrastructure) community responses showed considerable similarity to medieval practices. Gil-

Guirado et al. (2016) proposed the Perceptual Index for Changes in Climate Risk (PICCR) and identified four phases in societal responses to extreme events such as floods and droughts, presented and discussed on the basis of a multi-centennial Spanish database. The first phase, covering the 17th and the mid-18th centuries, had a passive religious approach and a lack of appropriate techniques, while the second phase, from the mid-19th to the mid-20th century, enjoyed some more advanced technologies, blaming natural processes for the danger. The third and fourth phases appear from the second half of the 20th century, when

applied technologies created a false sense of security and led to risk avoidance strategies in the fourth phase.

Adamson (2014) examined late 18th- and 19th-century responses to drought in western India at governmental level, using mainly English-language sources. In general, the society of the time was usually able to cope with droughts arising out of failures of monsoon rainfall over a large spatial extent: during the 1790–1860 study period; only 1803, when significant, large-scale famine developed in this region, stands out. Nonetheless, local famines in years with drought-driven harvest failure did occur. During

years with grain shortages, cultivators usually had to take more loans, steadily exacerbating their debt-bondage. This resulted, for example, in waves of population movement (e.g. in 1790, 1803, 1812, 1824, 1838) and social tensions (e.g. riots in 1832). Another form of response was to increase the areas of cultivated land, thus increasing the chances that more grain would come to market even in bad years. In fact, the system reflected the interests of grain dealers and/or moneylenders rather than those of society as a whole, and individuals were quite vulnerable to the negative consequences of climate-driven stress factors. Smithsonian political

economics was broadly applied as a framework by colonial society, ensuring the highest possible financial benefits to the government. However, governmental policies, such as banning grain export, proved ineffective against high mortality driven by famine in affected areas. Market-driven drought policy proved to be relatively successful (or at least managed to mask or buffer internal problems) in less significant droughts but rendered society even more vulnerable in the event of truly severe drought events. In a follow-up paper, Adamson (2016) discussed the socio-economic impacts and social responses to droughts in Western

India during the 1782–1857 period. Until the mid-19th century, agricultural communities (both farmers and pastoral people) were faced with high drought mortality. Apart from immediate survival strategies, such as the consumption of famine crops or migration to the towns, different responses developed at individual level, including changes in agricultural practices (e.g. rotation), rogation ceremonies, and – inevitably – negotiating more loans. Although charitable support on the part of rulers and the affluent classes was also common, colonial public support policies can be also traced in drought-threatened areas from 1817 onwards.

Campos (2015) investigated public policies and paradigms in north-eastern Brazil in historical perspective, covering a ~200-year period but also combining earlier drought-related documentary evidence from the great drought of 1583 onwards. As well as drought-related historical paradigms of hydraulic (state-organised) and ecological solutions, and economic development, the author also examines the evolution of administrative solutions, management and public policies, comparing the great droughts of 1877–1879, 1958, and 2012–2013.

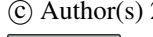


Berland and Endfield (2018) investigated the human and financial consequences, as well as crisis management, of a severe drought that occurred on the island of Antigua (Lesser Antilles) at the time of the American War of Independence (1775–1783), together with governmental responses to it. Although the many fatalities and grave economic losses of these years were resulted from, or were a combination of, many important factors, including geopolitical and socio-economic stress, drought played a

significant role in producing these problems.

Pribyl et al. (2018) presented and analysed the causes, main characteristics and short-term as well as long-term socio-economic consequences of the great drought of the 1890s in south-eastern Africa. The most significant short-term consequences were acute shortages and famine, combined with related environmental crisis factors, such as locust outbreaks and cattle plague, that led to a severe crisis in agriculture. The great drought and associated economic-environmental disaster catalysed the decline of

traditional agriculture and contributed to the movement of increasing numbers of indigenous people, forced to give up agriculture and become miners in the diamond industry. The paper also discussed the colonial administrative response, shortage management strategies and their effectiveness during the crisis.

According to Xiao et al. (2018), famines that arose after flood and drought extremes in Northern China during the late Quing dynasty (1790–1911), elicited public responses in the form of charity action, revolts (in the countryside) and crisis-related

public policies (in Beijing). Such calamities usually generated high numbers of refugees, who required further intensive governmental management. In an earlier paper, Xiao et al. (2014) examined different societal response strategies in the same region back to the beginning of the above dynasty in 1644.

**5 Discussion**

**5.1 The spatio-temporal variability of droughts**

As Sect. 4.1 shows, there is a wealth of literature examining the long-term fluctuations of droughts as revealed by documentary evidence from many parts of the world. Such studies differ in the density and quality of documentary data (from which droughts are identified) they contain, in their definitions and selection of droughts, in the areas analysed, as well as in the time periods covered. It is not surprising that drought chronologies are usually correlated with those from nearby areas. Moreover, in such comparisons, it must also be taken into account that forcings and circulation factors driving the occurrence of drought episodes

may be geographically variable. This section presents potential teleconnections reflected in the concurrent (or opposite) occurrence of drought-rich and drought-poor periods.

Fig. 9 shows a selection of long-term decadal fluctuations in several documentary-based drought characteristics related to Europe over the past millennium, extracted from a range of papers: (i) frequency of dry months in Western Europe from chronicle sources (Alexandre, 1987); (ii) standardised weighted index for drought rogation ceremonies for four sites (Barcelona, Girona,

Tarragona, Tortosa) on the Catalonian coast of north-eastern Iberia (Oliva et al., 2018); (iii) frequency of droughts and PDSI over the territory of the Czech Lands (Brázdil et al., 2013, 2016a); (iv) 30-year frequencies of extremely dry MAM–JJA seasons in the European non-chernozem part of the former Soviet Union (Lyakhov, 1984). While both the Czech series are highly correlated (r = –0.65, statistically significant at the 0.05 significance level), unsurprisingly, given their different climatic context, no relationship can be found between the Czech drought series and the drought rogation index of the Catalonian coast of north-eastern Iberia.

Turning to the long-term decadal fluctuations of documentary-based drought characteristics in China, Fig. 10 shows drought variability over the two past millennia for the following series: (i) decadal means of regional dry-wet index series for the North China plain (c. 34–40°N, east of 105°E), the Jiang-Huai area (c. 31–34°N, east of 110°E), the Jiang-Nan area (c. 25–31°N, east of 110°E) and (central) Eastern China (c. 25–40°N, east of 105°E), derived from an annual drought/flood grade dataset after





detrending the effects of absent data on homogeneity (Ge et al., 2014, 2016); (ii) decadal means of degrees of dryness/wetness in the Great Band of the Yellow River region (Yang et al., 2014); (iii) decadal frequencies of droughts in the Yangtze Delta (Jiang et al., 2005). As might be expected, dry-wet index series correlate better between neighbouring areas (r = 0.52 for North China versus the Jian-Huai area; 0.46 for Jian-Huai area versus the Jiang-Nan area), while for the two more distant regions the figure is only

0.18. The North China series express the best correlation with the mean Eastern China series (r = 0.81) and the worst (r = 0.62) with the Jiang-Nan area. The series for the Great Band of the Yellow River region is best correlated with North China (r = 0.67) and the series of droughts frequencies in the Yangtze Delta with the Jian-Huai area (r = −0.46). All these correlation coefficients are statistically significant at the 0.05 significance level.

Annually-resolved drought reconstructions for Africa are largely restricted to the 19th century. The various series for

southern Africa appear in Fig. 11, together with a rainfall reconstruction for Zimbabwe based on tree-ring variability (Therrell et al., 2006). Assessing any statistical relationships between these series is challenging, as some series are based on conditions during calendar years, while others consider the July–June hydrological year. However, when taken as a whole, the first decade of the 19th century appears very dry (although note the discrepancies between the documentary and tree-ring-/ships'-logbook-derived reconstructions, as discussed in Sect. 4.1.3). Other widespread drought episodes centre around the mid-1820s, the mid-1830s, the

late 1850s to mid-1860s, mid-late 1870s, early-mid 1880s, and mid-late 1890s (Nash, 2017). Individual studies suggest that, of these, the drought during the early 1860s was the most severe of the 19th century, with the droughts of the 1820s and 1890s the most protracted. It should, however, be noted that the spatial extent of individual drought episodes was highly variable. The drought of the late 1890s, for example, was severe over much of South Africa and Lesotho, but regions further north were apparently less affected.

Drought series available for the Americas relate largely to the past 500 years (Fig. 12). The following drought series are available: (i) decadal frequencies of droughts in central Mexico (Mendoza et al., 2005a); (ii) decadal frequencies of very dry and dry years on the Pacific coast of Central America (Guevara-Murua et al., 2018); (iii) decadal frequencies of dry and very dry years for Potosí, Bolivian Altiplano, Bolivia (Gioda and Prieto, 1999). The differences in the fluctuation of drought frequency in all three areas are reflected in their low and statistically non-significant correlation coefficients.

**5.2 Droughts as derived from documentary evidence and natural proxies**

Direct comparisons of drought series (wet and dry episodes) derived from documentary evidence with reconstructions based on natural proxies are relatively rare. The PAGES Hydro2k Consortium (2017) analysed hydroclimate variability and change spanning the Common Era, comparing natural proxies (corals, tree rings, speleothems and sediments) and model estimates. Documentary data were also reported in this paper, with particular emphasis on related studies in Europe and Asia.

Comparisons between hydroclimate reconstructions based on documentary evidence and tree-ring studies work well in regions where both types of data are available. Considerable numbers of such papers have appeared, for example, in central Europe, where close collaboration between historical climatologists and dendroclimatologists has been established. In particular, documentary data have been used in the confirmation of hydroclimatic extremes following from tree-ring reconstructions. As an illustration, Brázdil et al. (2002) compared information from documentary data for years with extremely thin (dry patterns) and

extremely wide (wet patterns) tree-rings of fir *Abies alba* in South Moravia (Czech Republic), which were used to reconstruct of March–July precipitation totals in the 1376–1996 period. Comparison of documentary data with extremes from a May–June Z-index series, reconstructed from an extended fir-tree-ring dataset in South Moravia for 1500–2008, was provided by Büntgen et al. (2011a). In similar fashion, Büntgen et al. (2010a) used comparisons of extremes in reconstructed PDSI series from pine *Pinus sylvestris* tree-rings in northern Slovakia for the 1744–2006 period. In the analysis of fir tree-rings from samples taken across





France, Switzerland, Germany, and the Czech Lands for the AD 962–2007 period, documentary data were used not only for confirmation of tree-ring extremes, but also in the development of documentary-based temperature and precipitation indices for Germany, the Czech Lands and Switzerland (Büntgen et al., 2011b). Later Dobrovolný et al. (2015b) compared documentary data with extremes detected in oak (*Quercus robur* or *Q. petraea*) tree-rings in the Czech Republic from AD 761 onwards.

Subsequently, Dobrovolný et al. (2018) used such comparisons for a May–July precipitation reconstruction from oak tree-rings for Bohemia (Czech Republic) from AD 1040 onwards.

Linderholm and Molin (2005) combined weather reports from a farmer's diary kept between 1815 and 1833 with SPI series calculated from Scots pine *Pinus sylvestris* tree-ring widths in east-central Sweden starting in 1751, and established the 1806–1835 drought as the longest continuous such event in the last 250 (or even 300) years. Matskovsky et al. (2017) used

documentary data in the analysis of hydroclimate variability since the 1790s in the Voronezh region (Russia) based on pine tree-rings. More recently, Sun et al. (2018) applied documentary evidence for comparison with droughts reconstructed from tree-rings of the purple-coned spruce *Picea purpurea* Mast from Mt. Shouyang in the source region of the Weihe River in north-western China from AD 1810 onwards.

Although the agreement between extremes derived from documentary and tree-ring data is generally good, there are some

discrepancies. Wetter et al. (2014) used the term "megadrought" to describe extensive, 11-month dry patterns documented in central and western Europe in 1540. This term has come to be generally applied to longer and more extensive drought events in other parts of the world (compare e.g., Stahle et al., 2007; Cook et al., 2010b). Moreover, Büntgen et al. (2015) argued that 1540 did not appear as a significant extreme year in many tree-ring chronologies from all parts of Europe. Pfister et al. (2015) replied, in support of the original concept, using further arguments to reinforce the documentary data from which the extreme dryness of 1540 had been derived. Extreme dry patterns in 1540 were later confirmed by the reconstruction of drought index series (clearly

had been derived. Extreme dry patterns in 1540 were later confirmed by the reconstruction of drought index series (clearly expressed in SPI, SPEI and Z-index, but not in PDSI) from documentary data by Brázdil et al. (2016a) and SPEI from grape-harvest dates by Možný et al. (2016b). Because summer PDSI is usually among the drought indicators reconstructed from tree-ring series (e.g. Büntgen et al., 2010a; Cook et al., 2015), this may explain the discrepancy.

Documentary-based reconstructions of drought indices in the Czech Lands from AD 1501 onwards (Brázdil et al., 2016a)

have also been compared with tree-ring based reconstructions. While correlation of drought index series with other precipitation series proved somewhat weak, they correlated significantly with hydric-sensitive tree-ring series (correlation coefficients between 0.4 and 0.5), that is with South Moravian series of March–July precipitation totals (Brázdil et al., 2002) and May–June Z-index (Büntgen et al., 2011a) derived from fir *Abies alba* tree-ring chronologies. Also, a European OWDA JJA scPDSI reconstruction by Cook et al. (2015) confirmed good agreement with corresponding Czech drought indices.

Yang et al. (2013) compared a dryness/wetness index derived from Chinese historical documents covering 120 sites (Academy of Chinese Meteorological Science, 1981) for AD 1470–2000 with the Monsoon Asia Drought Atlas (MADA) derived from 327 tree-ring series by Cook et al. (2010a). Yang et al. (2013) demonstrated that the MADA alone cannot effectively represent dryness and wetness in eastern China, probably because of the lack of proxy records used for MADA, especially in north-eastern China.

In Africa, relatively few opportunities exist for direct comparison of annually-resolved documentary drought reconstructions with those derived from natural proxies. The drought chronology produced by Nash et al. (2018) for Malawi (see Fig. 11) can be compared with the reconstruction of water levels in lakes Malawi and Chilwa in Nicholson (1998b). Her record discloses that Lake Malawi maintained a relatively high level from the late 1850s to the 1890s. The impacts of the droughts in the early 1860s, mid-late 1870s and, in particular, the mid-1880s are, however, clearly visible. The lake began to recede in the mid-

1870s, with a dramatic decline occurring sometime after the mid-1880s. Lake Chilwa has a much smaller catchment and is more





susceptible to local rainfall variability. This lake rose rapidly in the 1840s before falling in the 1860s, rose again in the mid-1870s before falling in the mid-1880s, and remained relatively dry for the rest of the 19th century. These fluctuations show close agreement with available documentary-derived drought series. The discrepancies between the tree-ring width series for Zimbabwe (Therrell et al., 2006) and the documentary/gauge drought series of Nicholson et al. (2012b) have been noted in Sect. 4.1.3.

Very few studies exist in which information derived from documentary data and natural proxies has been used to produce new precipitation reconstructions. Casty et al. (2005) applied principal component regression analysis to long instrumental series and documentary proxy evidence to reconstruct 500-year temperature and precipitation series for the European Alps. They identified 1540, 1921 and 2003 as probably the driest in the past 500 years. A gridded precipitation reconstruction for Europe by Pauling et al. (2006) for the 1500–2000 period combined documentary data with tree-rings, ice cores, corals and speleothems. This

series was then used for the study of winter precipitation trends over the past 500 years in south-western Norway and southern Spain/northern Morocco by Matti et al. (2009). Neukom et al. (2010) used a multi-proxy approach (including documentary evidence) to provide the first reconstruction of multi-centennial precipitation variability for southern South America to be accurately gridded as well as highly resolved in spatial and temporal terms, covering the past half-millennium. They did not specifically address drought but reported on drier regions and periods. Their gridded products pave the way for future, high-

resolution, spatio-temporal drought analysis. Shi et al (2017) used 371 tree-ring and isotope series together with 107 documentary-based drought/flood indices to reconstruct the precipitation field over China for the past 500 years. Multiproxy reconstructions for southern Africa by Neukom et al. (2014) and Nash et al. (2016b) have been noted in Sect. 4.2.

## 6 Conclusion and future prospects

This study provides a worldwide review of research papers to date, published predominantly in international, English-language,

peer-reviewed journals, related to droughts derived from documentary evidence. It has included consideration of their spatio-temporal variability, forcings, socio-economic impacts and societal responses. The main points arising from the review may be summarised as follows:

(i) Documentary evidence offers a broad variety of sources types from which high-resolution data concerning droughts and related phenomena can be extracted. There appear to be no limits on identifying dry episodes in any particular part of the year from such

sources. The character of documentary data allows various types of drought to be identified, including the meteorological, hydrological, agricultural and socio-economic categories.

(ii) Droughts identified from documentary evidence during the pre-instrumental period are of key importance in world-wide spatio-temporal analyses of the fluctuations, teleconnections, seasonality, forcings and impacts of drier periods (particularly extreme events). Documentary data on droughts during the instrumental period are important to demonstrate the human impacts and

responses of former societies.

(iii) Compiling drought-related information from documentary data and meteorological/hydrological measurements (applying standard palaeoclimatic reconstruction approaches) is a necessary step in the evaluation of the importance of documentary data for the study of the long-term spatio-temporal variability of droughts. Co-operation between historical climatologists and the dendroclimatological community is vital for (a) further improving drought knowledge and (b) evaluating the advantages/drawbacks

of documentary and tree-ring data in their studies.

(iv) Series of documentary-based droughts reflect the effect of the circulation patterns typical of a given area. Particularly important relationships appear between droughts and ENSO in areas sensitive to these effects. Other natural forcings, including the possible effects of solar factors and/or volcanic activity, have also been reported. Anthropogenic forcing on droughts appears ongoing, largely under the influence of increasing temperatures due to increasing concentrations of greenhouse gases.





(v) Documentary evidence provides valuable information concerning the direct and indirect impacts of droughts on society, and societal responses to the environmental stresses brought on by drought. Historical documents are the best source for detecting and understanding societal vulnerability to drought, as well as the chief source of information for detecting how individuals, communities and local/regional/state administrations have responded to drought-related crises.

Despite a large number of papers addressing various aspects of drought derived from documentary data, more concentrated efforts in the following areas could contribute to further improvements in knowledge:

(i) The creation of more long-term series of droughts by combining high-resolution documentary data from the pre-instrumental period with droughts derived from meteorological measurements in the instrumental period, in order to demonstrate long-term spatio-temporal changes influenced by both natural and anthropogenic forcings.

(ii) Reconstruction of series of selected drought indices for the study of long-term variability and trends in regions where temperature and precipitation reconstructions are already available.

(iii) Comprehensive analysis of past major drought events from meteorological and/or climatological perspectives and examination of any associated human impacts and responses, thus providing estimates of the potential impacts of extreme droughts for current and future societies.

(iv) Collaboration with other specialists working with high-resolution proxy data (particularly tree-rings) in order to improve understanding of past droughts by combining and confronting various types of drought information (direct and proxy).

(v) Communication with climate modellers about methods of employing drought reconstructions in past model simulations and in the study of drought forcings, as well informing them as to how knowledge of the past can be used in future modelling projections of droughts.

(vi) Collaboration with (hydro-)climate reconstruction experts, economic and environmental historians, anthropo-geographers and other social scientists in order to provide the most comprehensive information possible concerning what lessons for the future may be learnt from successful and unsuccessful drought management practices in the past.

**Data availability.** Drought series used in the paper are available from the corresponding authors.

**Competing interests.** The authors declare that they have no conflict of interest.

**Acknowledgements.** Rudolf Brázdil and Ladislava Řezníčková acknowledge the support of Czech Science Foundation, project no. 17-10026S, and the Ministry of Education, Youth and Sports of the Czech Republic within the National Sustainability Program I (NPU I), grant number LO1415. Jürg Luterbacher acknowledges support from the German Federal Ministry of Education and Research (BMBF) and JPI-Climate/Belmont Forum collaborative Research Action "INTEGRATE, An integrated data-model study of interactions between tropical monsoons and extratropical climate variability and extremes". David J. Nash's contributions were 30   supported in part by the Leverhulme Trust Research Project, Grant number F/00 504/D. Kunhai Elaine Lin (Taipei, Taiwan) is acknowledged for information related to the database of Chinese historical-climatological data. We thank the following colleagues for kindly providing us with their drought series: Maria del Rosaria Prieto (Mendoza), Mariano Barriendos (Barcelona), Bao Yang (Lanzhou), Facundo Rojas (Mendoza) and Jingyun Zheng (Beijing). We would also like to thank the National Drought Mitigation Center, University of Nebraska-Lincoln for copyright clearance for Fig. 1, the Bibliothèque Nationale de France (Paris) for Fig. 3, 35   Oldřich Kotyza (Litoměřice) for Fig. 4, Muzeum Narodowe w Warszawie for Fig. 8, and Tony Long (Svinošice) for English style corrections.





## Archival sources

[AS1] Archives of the Vereinte Evangelische Mission Wuppertal, catalogue no. RMG 2.585 C/i 6, 209: Okahandja, Otjikango (Neu-Barmen), Otjimbingue u.a.m.: Beiträge zur Geschichte der Stationen. 1848–1892.

[AS2] Arxiu Capitular de la Cathedral de Barcelona, "Exemplaria", vol. V, fol. 96, 3/3/1691.

[AS3] Bibliothèque Nationale de France, Mexicain 385: Codex Telleriano-Remensis.

[AS4] Klíč od deště aneb Nová píseň v čas sucha. Prosba a klíč od deště, též za odvrácení hladu, moru, vojny i jiné potřeby celého křesťanstva ke cti Pánu Bohu Hospodáři Nebeskému, jejž on v své moci má, též dešťové nejsvětější Panně Marii Matce Boží Vyšehradské a svatým patronům českým na den svatého Václava léta 1678, když po velikém suchu spadla z nebe ponejprv vděčná vláha, na poděkování toho i posavád trvajícího božího dobrodiní od Václava Šťastnýho Františka Rambeka, měštěnína N[ového]
M[ěsta] P[ražského] k zpívání přihotovený. J[iž po] druhé vytištěný v Starém Městě Pražském u Daniele Michálka léta 1679.

[AS5] Moravská zemská knihovna Brno, sign. A21: Hieronymus Haura, Miscellanea iucundo-curiosa in quibus continentur variae descriptiones, versus, carmina, elogia, epitaphia, vaticinia, illuminationes, declarationes, pugnae, conflictus, notata de bellis et diversis temporibus, casus laeto-fatales, contingentia in monasterio Sancti Thomae, processiones et devotiones ad Thaumaturgam, varii eventus in Moravia, Bohemia, et adjacentibus regionibus, Brunae et aliis civitatibus, ac aliae iucundae, et utiles annotationes
et reflexiones ... Quae omnia diligenter annotavit, laboriose conscripsit Pater Hieronymus Haura, Boemus Moldavo-Teynensis, Ord. Erem. D. P. Augustini, Brunae in Exempto Monasterio S. Thomae Professus ... T. III.

[AS6] Moravský zemský archiv Brno, fond B13 Moravské místodržitelství – presidium 1850–1918, inv. č. 523, sign. 1/19: Požáry, záplavy 1904–1905.

[AS7] Moravský zemský archiv Brno, fond E55 Premonstráti Hradisko, sig. II–29: Diaria kanonie Klášterní Hradisko 1693–1783.

[AS8] Státní okresní archiv Litoměřice, fond AM Litoměřice, st. sign. 12: Letopisecké záznamy v litoměřickém právním rukopise ze 14. stol. označený nově "Das Magdeburger Recht" [1426–1574].

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





**Table 1.** Indicators of individual types of drought in documentary evidence.

| Type of drought | Documentary indicators |
|---|---|
| meteorological | Accounts of a lack of rain, drought, dry weather, hot and dry weather, periods without rain, dust on roads, rain that hardly moistened the soil, rain needed, "beyond-living-memory" drought (dryness) |
| agricultural | Accounts of complete failure of crops or bad harvest, lack of seed, lack of feed for livestock, cracked earth, dried-out pastures, limited availability of straw, conditions that are impossible for soil cultivation or sowing, tearing of grain by hand rather than reaping, caterpillars and other pests, damage to crops |
| hydrological | Accounts of the appearance of hunger stones, low water levels in rivers, standing and/or green water in rivers, crossing of otherwise large rivers "barefoot" or with wagons, drying out of springs, wells, fountains, brooks, streams and fish cultivation ponds, lack of water for people and animals, sale of water, water-mills out of operation, cessation of river transport, lack of water for extinguishing fires. [N.B. Many of these indicators require careful interpretation in dry-climate regions of the world where dry seasons are a common part of the hydrological pattern.] |
| socio-economic | Accounts of food shortages, increase of prices (grain and other crops), poverty, debt, distress, famine, requests for tax reduction, administrative measures, raised awareness of witchcraft and other rain-related ritual practices, human mortality, disease, epidemics, emigration, building and forest fires, sale of livestock at below normal market prices |





**Figure 1.** Sequence of drought occurrence and impacts for commonly accepted drought types. Source: http://drought.unl.edu/Education/DroughtIn-depth/TypesofDrought.aspx, last access: 27 August 2018. Copyright: National Drought Mitigation Center, University of Nebraska-Lincoln, U.S.A.





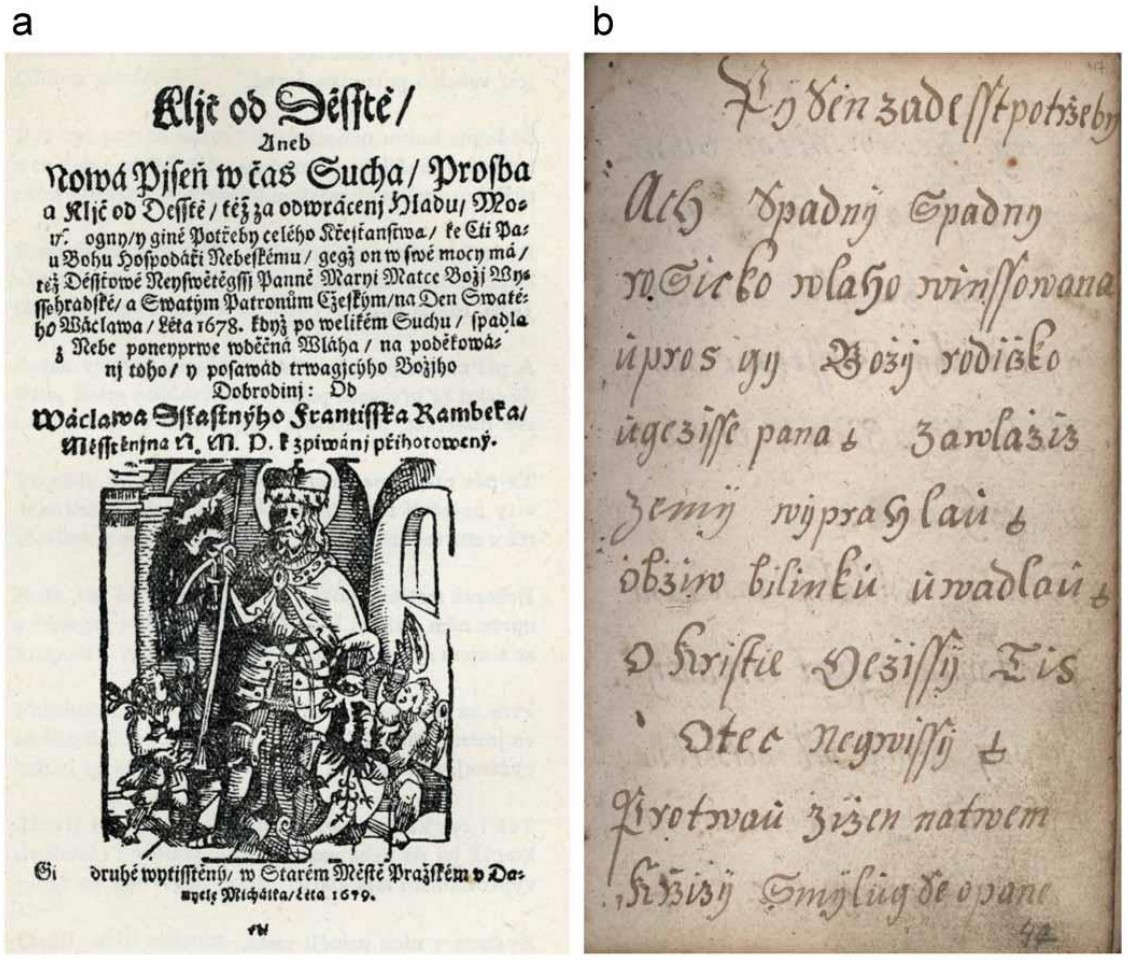

**Figure 2.** a) Title page of "Key to the Rain, or a New Song for a Time of Drought", second edition, published in Prague in 1679 (AS4); b) Introductory page of "Song in Need of Rain" from manuscript records of Antonín Štěpán, a citizen of Pelhřimov (Martínková, 2005).

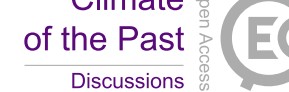

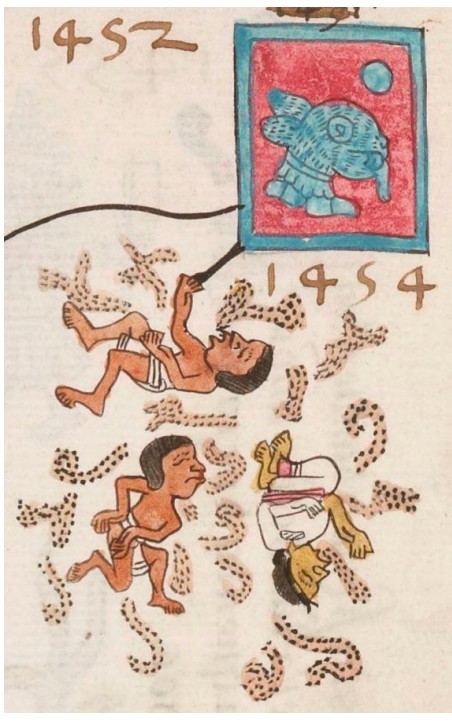

**Figure 3.** Expression of dust storms and people who succumbed to famine, from the *Codex Telleriano-Remensis* (AS3), folio 32v(erso), portraying the Famine of the year of The One Rabbit in 1454 (year sign for One Rabbit, top right). According to Therrell et al. (2004), the famine resulted from a multi-year drought, possibly coupled with an early autumn frost in 1453.





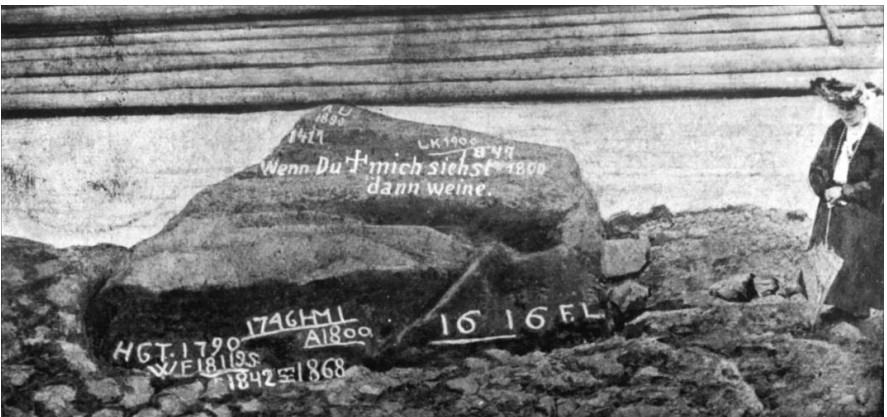

**Figure 4.** A hunger stone that appeared during the severe 1904 drought, situated on the left bank of the River Elbe at Děčín-Podmokly, with records of years and low water levels and a warning inscription (O. Kotyza archive).



**Figure 5.** Various types of long-term drought reconstructions for the Czech Lands: a) April–September sums of precipitation indices from documentary data for 1501–1854 and precipitation totals for 1855–2014; b) decadal frequencies of drought episodes for 1501–2012 (Brázdil et al., 2013); c) April–September SPEI and Z-index for 1501–2014 derived from documentary data (Brázdil et al., 2016a); d) April–August SPEI for 1499–2012 from grape harvest dates (Možný et al., 2016b). Smoothed by Gaussian filter over 20 years (black curves).



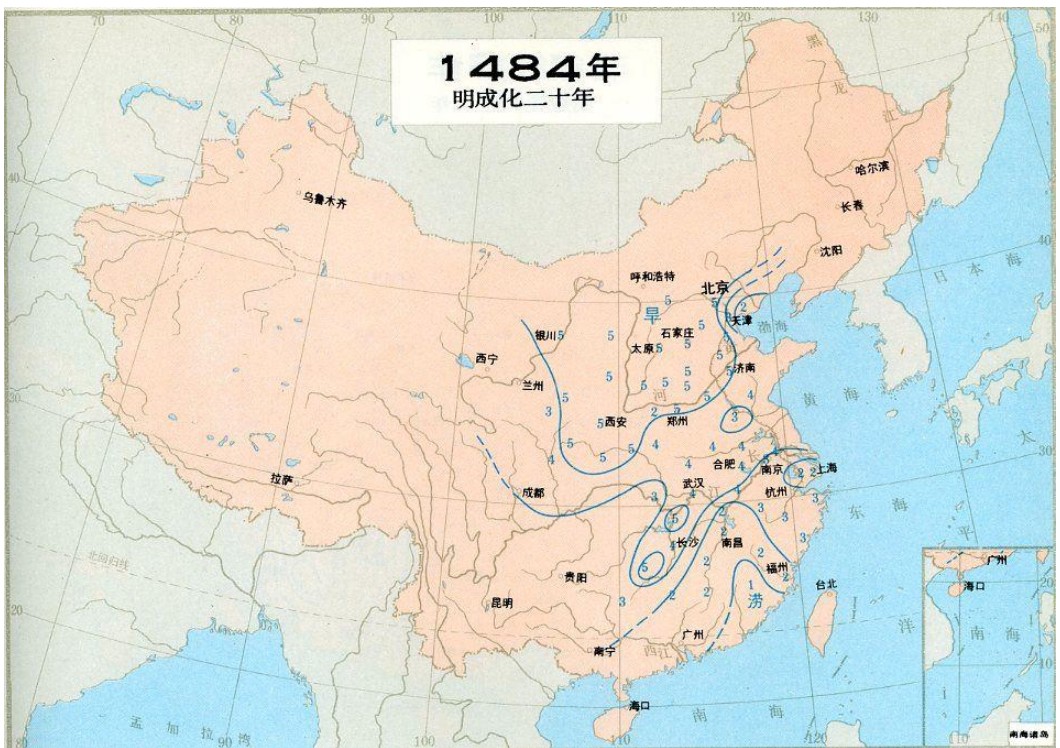

**Figure 6.** Annual dryness/wetness indices in China in AD 1484: 1 – very wet, 2 – wet, 3 – normal, 4 – dry, 5 – very dry (adapted, after the Academy of Chinese Meteorological Science, 1981).



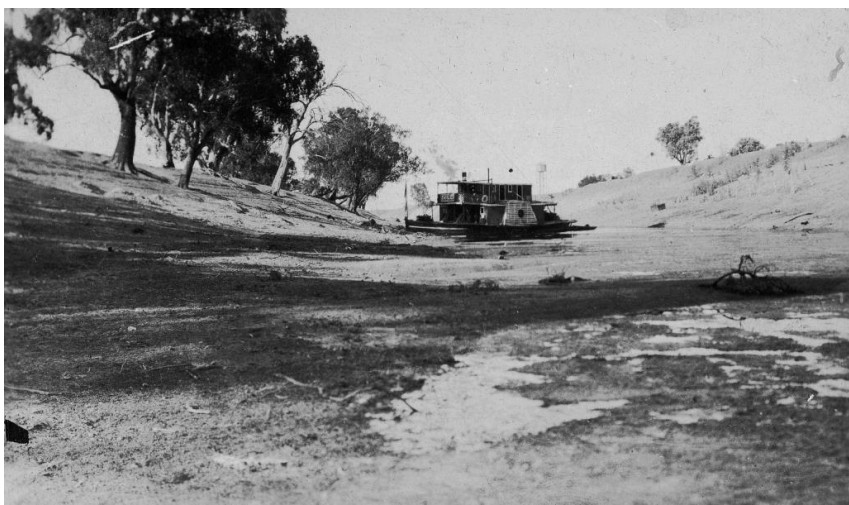

**Figure 7.** Photographic postcard showing the paddle-steamer S. S. Nile on the bed of the Darling River south of Bourke (New South Wales, Australia) during a drought. The image was taken by the journalist C.E.W. Bean for the Sydney Morning Herald, probably in 1908 or 1909, in the course of his two trips to New South Wales (Australian National Maritime Museum on The Commons, Object no. 00017014). Gergis and

5    Ashcroft (2012) identified all years between 1904 and 1909 as dry for this area.





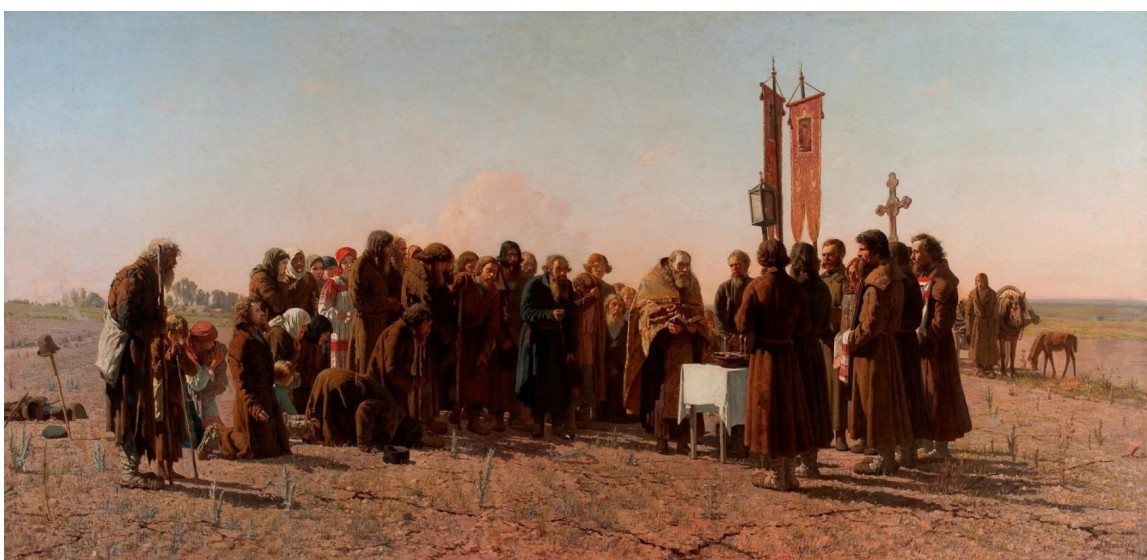

**Figure 8.** "Prayer in Time of Drought" by the Russian artist Grigoryi Grigorievich Myasoyedov (1834–1911, this work dating to between 1878 and 1881) shows poor people praying for rain, a traditional response of to drought (Muzeum Narodowe w Warszawie, M.Ob.2365; Miasojedow Grzegorz G. (malarz); *Prayer in Time of Drought*; 1878; oil; canvas; 140x285,5; photo: K. Wilczyński).




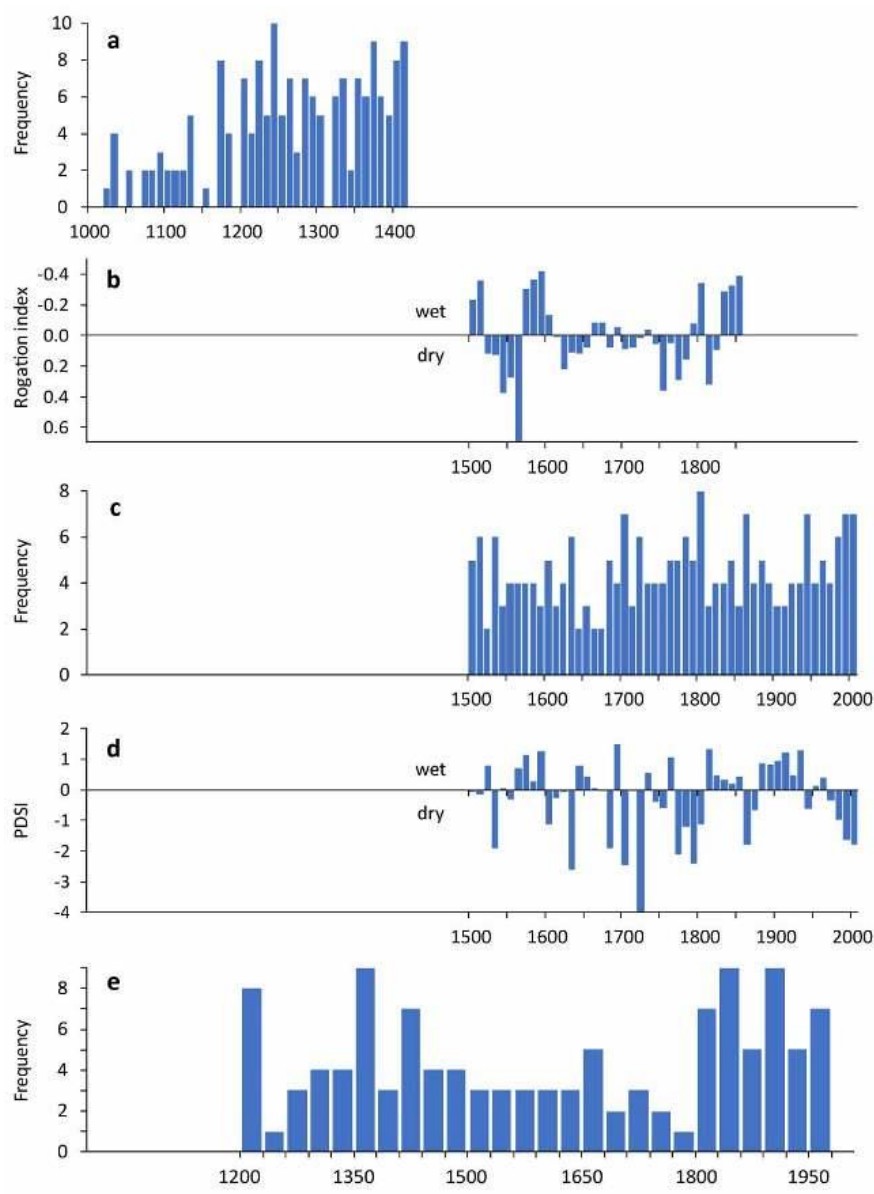

**Figure 9.** Long-term fluctuations in a range of documentary-based drought series across Europe: a) decadal frequencies of unusually dry JJA months in Western Europe, 1000–1419 (Alexandre, 1987); b) decadal means of the weighted drought rogation index for the Catalonian coast of north-eastern Iberia, 1501–1860 (Oliva et al., 2018); c) decadal frequencies of droughts in the Czech Lands, 1501–2010 (Brázdil et al., 2013); d) decadal means of annual PDSI in the Czech Lands, 1501–2010 (Brázdil et al., 2016a); e) 30-year frequencies of extremely dry MAM–JJA seasons in the European non-chernozem part of the former Soviet Union, 1201–1980 (Lyakhov, 1984).





**Figure 10.** Long-term fluctuations in a range of documentary-based drought series across China: a–d) decadal means of regional dry-wet index series for the North China plain (a), Jiang-Huai area (b), Jiang-Nan area (c), and (central) Eastern China (d), past 2000 years with missing or incomplete decades marked by points (Ge et al., 2014, 2016); e) decadal means of degrees of dryness/wetness (2 – wet, 3 – normal, 4 – dry – see Sect. 4.1.2) in the Great Band of the Yellow River region, 1470–2000 (Yang et al., 2014); f) decadal frequencies of droughts in the Yangtze Delta, 1000–2000 (Jiang et al., 2005).



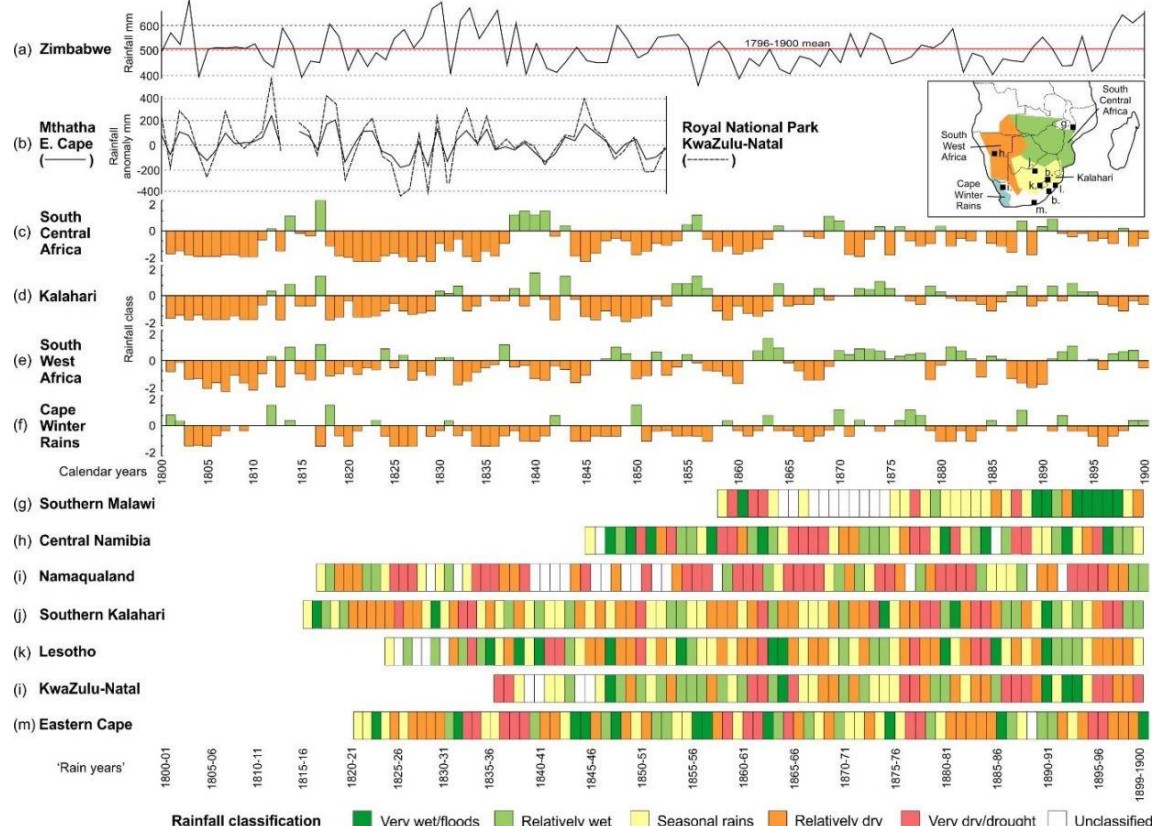

**Figure 11.** Long-term fluctuations in a range of documentary-based drought series for the southern African summer and winter rainfall zones for the 19th century (see Sect. 4.1.3): (a) tree-ring series for Zimbabwe (Therrell et al., 2006); (b) ships'-logbook-derived series for Mthatha, Eastern Cape, and Royal National Park, KwaZulu-Natal (anomaly relative to 1979–2008 mean) (Hannaford et al., 2015); (c–f) documentary- and gauge-data-derived composite series for "South Central Africa" (Zambia, northeast Botswana, Zimbabwe, southern Mozambique), the "Kalahari" (southern Botswana, northern/eastern South Africa), "South West Africa" (south-eastern Angola, Namibia, north-western South Africa), and "Cape Winter Rains" (south-western South Africa) (Nicholson et al., 2012b), (g–m) documentary-derived series for southern Malawi (Nash et al., 2018), central Namibia (Grab and Zumthurm, 2018), Namaqualand (Kelso and Vogel, 2007), southern Kalahari (Nash and Endfield, 2002a, 2008), Lesotho (Nash and Grab, 2010), KwaZulu-Natal (Nash et al., 2016b) and Eastern Cape (Vogel, 1989). Inset map (top right) shows areas covered by composites c–f, and the approximate locations of other series.



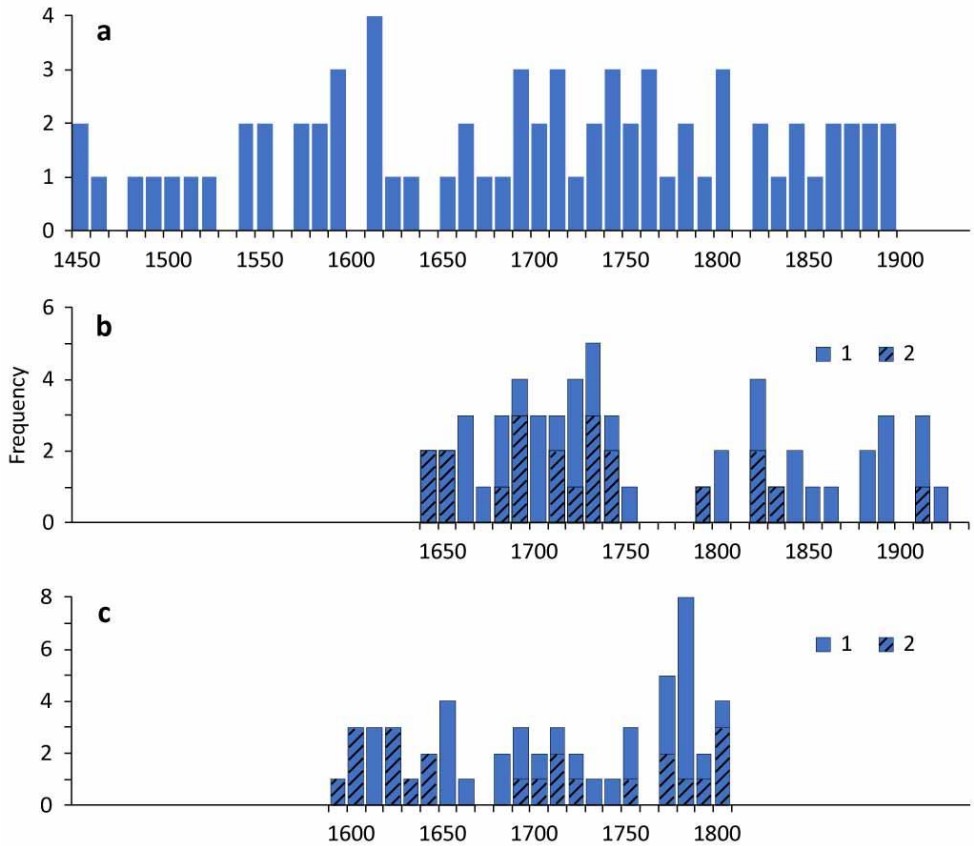

**Figure 12.** Long-term fluctuations in a range of documentary-based drought series across North and Central America: a) decadal frequencies of droughts in central Mexico, 1450–1900 (Mendoza et al., 2005a); b) decadal frequencies of droughts (1 – dry, 2 – very dry) of the Pacific coast of Central America, 1640–1945 (Guevara-Murua et al., 2018); c) decadal frequencies of droughts (1 – dry, 2 – very dry) for Potosí, Bolivian Altiplano, Bolivia, 1585–1815 (Gioda and Prieto, 1999).