# Peer review of "Documentary data and the study of past droughts: a global state-of-theart"

_Climate of the Past, 2018_

## Short Comment (SC1) · 28 Sep 2018

Dr. Van Lanen

henny.vanlanen@wur.nl

Dear authors,

I believe you missed some relevant references about studies on past drought, mostly from Europe. A synthesis can be found in: Stahl, K., Tallaksen, L.M. and Hannaford, J. (2018): Ch. 1.2 Recent Trends in Historical Drought, In: Iglesias, A., Assimacopoulos, D. and Van Lanen, H.A.J. (Eds.) (2018): Drought: Science and Policy. Wiley Blackwell, 258 pg. (https://www.wiley.com/en-nl/Drought:+Science+and+Policy-p-9781119017073)

I suggest to read the following: BARD, A., RENARD, B., LANG, M., GIUNTOLI, I., KORCK, J., KOBOLTSCHNIG, G., JANZA, M, D'AMICO, M., AND VOLKEN, D. (2015).

Trends in the hydrologic regime of Alpine rivers. Journal of Hydrology, 529, 1823-1837. Doi: 10.1016/j.jhydrol.2015.07.052 COCH A., AND MEDIERO, L. (2016) Trends in low flows in Spain in the period 1949–2009. Hydrological Sciences Journal, 21: 568-584. 10.1080/02626667.2015.1081202 DAI, A. Increasing Drought Under Global Warming In Observations And Models. Nature Clim. Change 3, 52–58 (2013). FOLLAND, C. K., HANNAFORD, J., BLOOMFIELD, J. P., KENDON, M., SVENSSON, C., MARCHANT, B. P., PRIOR, J., AND WALLACE, E. (2015) Multi-annual droughts in the English lowlands: a review of their characteristics and climate drivers in the winter half-year, Hydrol. Earth Syst. Sci., 19, 2353-2375, doi:10.5194/hess-19-2353-2015. GUDMUNSSON, L., AND SENEVIRATNE, S. I. (2015) European Drought Trends. IAHS, 369, 75-79. HANNAFORD, J., AND MARSH, T. J. (2006) an assessment of trends in UK runoff and low flows using a network of undisturbed catchments, Int. J. Climatol. 26, 1237–1253. DOI: 10.1002/JOC.1303 HANNAFORD, J., BUYS, G., STAHL, K., AND TALLAKSEN, L.M. (2013) The influence of decadal-scale variability on trends in long European streamflow records, Hydrol. Earth Syst. Sci., 17,2717-2733, DOI:10.5194/HESS-17-2717-2013. KINGSTON, D.G., STAGGE, J.H., TALLAKSEN, L.M., AND HANNAH, D.M. (2015) European-scale drought: understanding connections between atmospheric circulation and meteorological drought indices. J. Climate, 28, 505–516. DOI: 10.1175/JCLI-D-14-00001.1 NASR A., AND BRUEN M. (2017). Detection of trends in the 7- day sustained low-flow time series of Irish rivers, Hydrological Sciences Journal, DOI: 10.1080/02626667.2016.1266361 ORLOWSKY, B., AND SENEVIRATNE, S.I. (2013). Elusive Drought: Uncertainty In Observed Trends And Short- And Long-Term Cmip5 Projections, Hydrol. Earth Syst. Sci., 17, 1765–1781. SHEFFIELD, J., WOOD, E.F. AND RODERICK, M.L. (2012). Little Change In Global Drought Over The Past 60 Years. Nature 491, 435–438. SPINONI, J., NAUMANN, G., AND VOGT, J. (2015). Spatial Patterns Of European Droughts Under A Moderate Emission Scenario. Adv. Sci. Res., 12, 179-186. SPINONI, J., NAUMANN, G., AND VOGT, J.V. (2017). Pan-European seasonal trends and recent changes of drought frequency and severity. Global and Planetary Change, 148, 113-130. DOI:

10.1016/J.GLOPLACHA.2016.11.013 STAGGE, J.H., KINGSTON, D., TALLAKSEN, L.M., AND HANNAH, D. (2016). Diverging Trends Between Meteorological Drought Indices (SPI and SPEI). Geophysical Research Abstracts, VOL. 18, EGU2016-10703-1, 2016. STAHL, K., HISDAL, H., HANNAFORD, J., TALLAKSEN, L.M., VAN LANEN, H.A.J., SAUQUET, E., DEMUTH, S., FENDEKOVA M., AND JODAR, J. (2010). Streamflow trends in Europe: evidence from a dataset of near-natural catchments, Hydrol. Earth Syst. Sci. 14. p. 2367–2382. STAHL K., TALLAKSEN L.M., HANNAFORD J., AND VAN LANEN H.A.J. (2012). Filling the white space on maps of European runoff trends: estimates from a multi-model ensemble. Hydrol. Earth Syst. Sci. Discuss. 9. p. 2005–2032. STAHL, K., VIDAL, J.P., HANNAFORD, J., PRUDHOMME, C., LAAHA, G., TALLAKSEN, L. (2014) Synthesizing changes in low flows from observations and models across scales. In Daniell, T.M., Van Lanen, H.A.J., Demuth, S., Laaha, G., Servat, E., Mahe, G., Boyer, J-F, Paturel, J-E, Dezetter, A., Ruelland, D. (Eds.). Hydrology in a Changing World: Environmental and Human Dimensions. 30-35. IAHS Publ. No. 363. Stahl, K., Kohn, I., Blauhut, V. , Urquijo, J., De Stefano, L., Acácio, V., Dias, S., Stagge, J. H., Tallaksen, L. M., Kampragou, E., Van Loon, A. F., Barker, L. J., Melsen, L. A., Bifulco, C., Musolino, D., de Carli, A. , Massarutto, A., Assimacopoulos, D., Van Lanen, H. A. J. (2016) Impacts of European drought events: insights from an international database of text-based reports Nat. Hazards Earth Syst. Sci., 16: 801-819. TALLAKSEN, L.M., STAGGE, J.H., STAHL, K., GUDMUNDSSON, L., ORTH, R., SENEVIRATNE, S.I., VAN LOON, A.F. AND VAN LANEN, H.A.J. (2015): Characteristics and drivers of drought in Europe – a summary of the DROUGHT-R&SPI project. In: Andreu, J. et al. (Eds.) Drought: Research and Science-Policy Interfacing. CRC/Balkema Publishers. VICENTE-SERRANO, S. M, LOPEZ-MORENO, J.I., BEGUERIA, S., LORENZO-LACRUZ, J., SANCHEZ-LORENZO, A., GARCIA-RUIZ, J.M., AZORIN-MOLINA, C., MORAN-TEJEDA, E., REVUELTO, J., TRIGO, R., (2014) Evidence of increasing drought severity caused by temperature rise in southern Europe. Environmental Research Letters, 9, 044001. doi:10.1088/1748-9326/9/4/044001.

---

## Referee Comment (RC1) · Anonymous Referee #1 · 10 Oct 2018

This is a comprehensive review that covers an important area within historical climatology. Whilst it does not contain any new data it goes over a lot of ground and I think will be a useful benchmark for the state of the discipline, hopefully encouraging more research in this area. The synthesis of information from different parts of the world is particularly helpful.

My comments are mostly relatively minor, although there are a number of them as this review goes over a lot of ground. The only relatively major change I would like to see is much more attention paid to the relationship between spatial variability on droughts and forcings, as this could say something quite interesting about long-term changes in climatologies. As it stands section 5.1 is pretty short and is not integrated with section 4.3. You can probably lose some detail elsewhere if the word count is an issue.

[Figure]

Abstract: Are historical-climatological databases really a source?

p. 1 line 30 - Link between drought and desertification could be made a little clearer. I'm not sure it is enough just to say that they are 'related' – a period of droughts does not necessarily imply desertification.

p. 1 line 33 - Give more detail on the 4 types of droughts.

P. 2 line 29 – why only reference the Africa consortium?

Section 2.1 – you can probably lose some of the quotes here which get a little repetitive. A little more detail on the types on information included would be better.

Section 2.3 – also too much focus on quotations and not enough information on what economic records would tell us that others wouldn't. I imagine these would be quite unique, so some specific detail would be helpful.

Section 2.7 – more detail is needed here. When were these songs collected? How do we know whether the dates are correct? Given that they are in the folk tradition, how do we know that the text has not changed over time?

Section 2.8 – again, it would be better to discuss the types of descriptions that are found in newspapers than just to give a long quotation. Newspapers are of course exceptionally diverse, but are there any commonalities in descriptions of drought impacts?

Section 2.9 – could you provide a little more information on the way that these 'winter counts' were recorded?

Section 2.10 – a little more detail would be helpful here. Where and when were these records collected? Why were they expressed in this way?

Section 3 (iii) – any examples from outside of the Czech Lands?

Page 20 line 36 – personal bugbear here. It is fine to say that the 'Southern Oscillation

(SO) made only minor contributions to central European drought variability', but not the 'Southern Oscillation Index'. The index is just a set of numbers that represents the climatic forecing: the Southern Oscillation or ENSO. Suggest you replace SOI with SO throughout the text.

I'm a little unsure about 'positive values of the NAO had a strong effect on a December–August drought index' as well, but at least here you are associating an index with another index, rather than an index with a drought.

Page 24 line 12-25 – I'm not sure about this paragraph and I think you should re-move it entirely. There's nothing particularly significant about using documents to look at droughts during the 'instrumental period'; it may be unusual within historical clima-tology but there's a huge literature on 20th century droughts from outside of historical climatology, and the instrumental record itself of course tells us very little about drought impacts. You'd either have to cite a far larger literature than you do here (Sen? Muk-erjee? McAlpin? – and that is only India) or stick to what historical climatology can contribute, i.e. pre-instrumental.

Section 4.4 – I wonder if there is a way to restructure this section so that you don't end up with a subsection on 'ancient civilisations and colonial regimes'. I don't quite see the association between the Neo-Assyrian Empire, colonial Mexico and colonial Australia. I suggest removing this section altogether. The first and third paragraphs and the material on the Pueblo could go into section 4.4.1 and the material on Mexico into section 4.4.4.

Section 5.1 – I would like to see more of a discussion of the spatial extent of droughts in relation to the climatic forcings discussed in section 4.3. This is where documentary evidence could be particularly useful, for example to discuss how teleconnection pat-terns have changed over time. I know that Nash has done this a few times for southern Africa, for example.

Page 28 lines 32-34 – this analysis seems incomplete. What about the relationship

between the Czech series and the Soviet Union, or Spain and western Europe? Why might some of these series be related but others not?

Section 5.2 – The Cook et al. megadrought reconstructions should be mentioned here in comparison with documentary data. That team have done at least three now (North America, Monsoon Asia and Europe). Given that they're calculated only using tree rings and statistical analysis, documentary data could provide a useful verification or comparison.

---

## Referee Comment (RC2) · Anonymous Referee #2 · 12 Oct 2018

This extensive manuscript provides a thorough overview of historical drought studies based on documentary evidence for most of the world. The focus is more on Europe and Africa than other areas, but that makes sense given the author list. The article is very long but well written, and I recommend that it be published once the authors have considered the following points:

Major comments: - There's no real discussion in this article about the limitations of documentary sources when it comes to climatological analysis - the impact of non-climatic elements in many of the evidence types mentioned in section 2, and the subjective nature of human memory. Will this be covered in a separate article in the special issue? If not, I think it is worth adding a paragraph or two about it in the discussion. If there will be another article focussed on this, please mention it.

[Figure]

- It's not clear to me why the Americas and Australia are grouped together in section 4.1 and 4.3. Drought responses there are almost anti correlated, particularly rainfall variability driven by ENSO. Is it for colonial reasons, or because similar methods are used? If so, please state that explicitly. You also mention New Zealand in section 4.4.3 (page 26), but this is not included elsewhere. It would be better to split America and Australia into two separate parts: The Americas, and the South Pacific, including Australia, NZ and possibly Pacific Island studies (if there are any).

-Somewhere in section 2, I wonder if you could mention information that falls between documentary and instrumental. Things like counts of rain days derived from weather journals and newspapers, or crop yield totals. I don't think this needs its own section, but could be slotted into others to show that it's not only words that can be useful.

-Section 4.2 confused me a bit. Are these the biggest events to be found in the many paper listed in section 4.1, or large overarching droughts that affected many countries? I think it's the latter, but this could be clarified with an introductory sentence or two, or by reshaping the section to focus on the timing of events. You could even tighten this section, removing reference to droughts that only occur in one country.

Minor things: -Page 2, line 38: I don't think you need 'the' before Climate of the Past

-Page 4, line 32: 'Related legal trials' instead of 'Legal trials related'

-Page 4, line 33: 'fashions' rather than 'fashion' I think

-Page 9, line 27: using numbers, hyphens and minus signs together is confusing, can you use an equal sign or colon instead?

-Page 13, line 9: As above

-Page 13, line 27: 'Droughts were more extreme in these centuries than in the 20th century'. This sentence could be clarified.

-Section 4.1.2: No mention of Japan?

[Figure]

-Page 15, line 6: The term Nilometer could be explained better

-Page 18, line 21: 'most severe' rather than 'severest' I think

-Page 20, line 32: 'above effects' is unclear to me

-Page 22, line 27: 'Ashcroft' rather than 'Aschcroft'

-Page 23, line 14: thank you for teaching me the word 'transhumant', I've never seen it before

-Page 24, line 12: add 'the' before 'instrumental period'

-Section 4.4.1, final paragraph: I like this qualifier, and am sure there are many other sources of information about the pros and cons of environmental determinism. Perhaps the authors could provide another overarching reference to this topic?

-Page 25, line 13: errant bracket

-Page 25, line 36: 'most immediate impact' on what?

-Page 28, line 21-22: this sentence is a bit clunky and could be rearranged

-Page 28, line 24: add 'well' before correlated

-Page 31, line 23: remove the 's' from sources

-Section 6: it sounds like another area for concerted effort is cross regional comparisons of historical droughts.

---

## Referee Comment (RC3) · Anonymous Referee #3 · 23 Oct 2018

The paper "Documentary data and the study of the past droughts: an overview of the state of the art worldwide" by R. Brazdil et al. aims at presenting the state of the art for spatial-temporal analyses of droughts derived from documentary evidence. It gives an excellent overview of the topic discussing types of documentary sources, methods for reconstructing droughts from them, long-term drought series and related forcings and impacts and in my opinion it will give a valuable contribution to the special issue in which it is going to be published. The overall quality of the paper is excellent: even though it is very long it does not contain unnecessary information as the length of the paper reflects the huge amount of work the authors performed to review the state of the art of past drought analyses worldwide. This huge amount of work is also clear from the long and complete list of references that is provided in the paper. This paper will

really be a milestone for anyone interested in past drought reconstruction. The paper is also well-organized which makes it easy to read in spite of the very large amount of information that is provided. I therefore think that the paper can be published in its present form and the only minor suggestion I give to the authors is to consider whether it is really necessary separating section 4.2 from section 4.1. The differences between the events discussed in these two sections are in fact not completely clear and the events discussed in sections 4.2 could also be moved to section 4.1.

---

## Author Comment (AC1) · 30 Oct 2018

Dr. Van Lanen henny.vanlanen@wur.nl

Dear authors, I believe you missed some relevant references about studies on past drought, mostly from Europe. A synthesis can be found in: Stahl, K., Tallaksen, L.M. and Hannaford, J. (2018): Ch. 1.2 Recent Trends in Historical Drought, In: Iglesias, A., Assimacopoulos, D. and Van Lanen, H.A.J. (Eds.) (2018): Drought: Science and Policy. Wiley Blackwell, 258 pg. (https://www.wiley.com/en-nl/Drought:+Science+and+Policy-p-9781119017073). I suggest to read the following:

BARD, A., RENARD, B., LANG, M., GIUNTOLI, I., KORCK, J., KOBOLTSCHNIG, G., JANZA, M, D'AMICO, M., AND VOLKEN, D. (2015). Trends in the hydrologic regime of Alpine rivers. Journal of Hydrology, 529, 1823-1837. Doi: 10.1016/j.jhydrol.2015.07.052 COCH A., AND MEDIERO, L. (2016) Trends in low flows in Spain in the period 1949–2009. Hydrological Sciences Journal, 21: 568-584. 10.1080/02626667.2015.1081202 DAI, A. Increasing Drought Under Global Warming In Observations And Models. Nature Clim. Change 3, 52–58 (2013). FOLLAND, C. K., HANNAFORD, J., BLOOMFIELD, J. P., KENDON, M., SVENSSON, C., MARCHANT, B. P., PRIOR, J., AND WALLACE, E. (2015) Multi-annual droughts in the English lowlands: a review of their characteristics and climate drivers in the winter half-year, Hydrol. Earth Syst. Sci., 19, 2353-2375, doi:10.5194/hess-19-2353-2015. GUDMUNSSON, L., AND SENEVIRATNE, S. I. (2015) European Drought Trends. IAHS, 369, 75-79. HANNAFORD, J., AND MARSH, T. J. (2006) an assessment of trends in UK runoff and low flows using a network of undisturbed catchments, Int. J. Climatol. 26, 1237–1253. DOI: 10.1002/JOC.1303 HANNAFORD, J., BUYS, G., STAHL, K., AND TALLAKSEN, L.M. (2013) The influence of decadal-scale variability on trends in long European streamflow records, Hydrol. Earth Syst. Sci., 17,2717-2733, DOI:10.5194/HESS-17-2717-2013. KINGSTON, D.G., STAGGE, J.H., TALLAKSEN, L.M., AND HANNAH, D.M. (2015) European-scale drought: understanding connections between atmospheric circulation and meteorological drought indices. J. Climate, 28, 505–516. DOI: 10.1175/JCLI-D-14-00001.1 NASR A., AND BRUEN M. (2017). Detection of trends in the 7- day sustained low-flow time series of Irish rivers, Hydrological Sciences Journal, DOI: 10.1080/02626667.2016.1266361 ORLOWSKY, B., AND SENEVIRATNE, S.I. (2013). Elusive Drought: Uncertainty In Observed Trends And Short- And Long-Term Cmip5 Projections, Hydrol. Earth Syst. Sci., 17, 1765–1781. SHEFFIELD, J., WOOD, E.F. AND RODERICK, M.L. (2012). Little Change In Global Drought Over The Past 60 Years. Nature 491, 435–438. SPINONI, J., NAUMANN, G., AND VOGT, J. (2015). Spatial Patterns Of European Droughts Under A Moderate Emission Scenario. Adv. Sci. Res., 12, 179-186. SPINONI, J., NAUMANN,

G., AND VOGT, J.V. (2017). Pan-European seasonal trends and recent changes of drought frequency and severity. Global and Planetary Change, 148, 113-130. DOI: 10.1016/J.GLOPLACHA.2016.11.013 STAGGE, J.H., KINGSTON, D., TALLAKSEN, L.M., AND HANNAH, D. (2016). Diverging Trends Between Meteorological Drought Indices (SPI and SPEI). Geophysical Research Abstracts, VOL. 18, EGU2016-10703-1, 2016. STAHL, K., HISDAL, H., HANNAFORD, J., TALLAKSEN, L.M., VAN LANEN, H.A.J., SAUQUET, E., DEMUTH, S., FENDEKOVA M., AND JODAR, J. (2010). Streamflow trends in Europe: evidence from a dataset of near-natural catchments, Hydrol. Earth Syst. Sci. 14. p. 2367–2382. STAHL K., TALLAKSEN L.M., HANNAFORD J., AND VAN LANEN H.A.J. (2012). Filling the white space on maps of European runoff trends: estimates from a multi-model ensemble. Hydrol. Earth Syst. Sci. Discuss. 9. p. 2005–2032. STAHL, K., VIDAL, J.P., HANNAFORD, J., PRUDHOMME, C., LAAHA, G., TALLAKSEN, L. (2014) Synthesizing changes in low flows from observations and models across scales. In Daniell, T.M., Van Lanen, H.A.J., Demuth, S., Laaha, G., Servat, E., Mahe, G., Boyer, J-F, Paturel, J-E, Dezetter, A., Ruelland, D. (Eds.). Hydrology in a Changing World: Environmental and Human Dimensions. 30-35. IAHS Publ. No. 363. Stahl, K., Kohn, I., Blauhut, V. , Urquijo, J., De Stefano, L., Acácio, V., Dias, S., Stagge, J. H., Tallaksen, L. M., Kampragou, E., Van Loon, A. F., Barker, L. J., Melsen, L. A., Bifulco, C., Musolino, D., de Carli, A. , Massarutto, A., Assimacopoulos, D., Van Lanen, H. A. J. (2016) Impacts of European drought events: insights from an international database of text-based reports Nat. Hazards Earth Syst. Sci., 16: 801-819. TALLAKSEN, L.M., STAGGE, J.H., STAHL, K., GUDMUNDSSON, L., ORTH, R., SENEVIRATNE, S.I., VAN LOON, A.F. AND VAN LANEN, H.A.J. (2015): Characteristics and drivers of drought in Europe – a summary of the DROUGHT-R&SPI project. In: Andreu, J. et al. (Eds.) Drought: Research and Science-Policy Interfacing. CRC/Balkema Publishers. VICENTE-SERRANO, S. M, LOPEZ-MORENO, J.I., BEGUERIA, S., LORENZO-LACRUZ, J., SANCHEZ-LORENZO, A., GARCIA-RUIZ, J.M., AZORIN-MOLINA, C., MORAN-TEJEDA, E., REVUELTO, J., TRIGO, R., (2014) Evidence of increasing drought severity caused by temperature rise in southern Europe.

Environmental Research Letters, 9, 044001. doi:10.1088/1748-9326/9/4/044001.

Dear Dr. Van Lanen, We would like to thank you very much for your interest in our paper in Climate of the Past Discussion. You provide a really comprehensive and exciting list of papers devoted to droughts from a hydrological point of view. These should surely be cited in a standard paper dealing with droughts based on instrumental data. However, that is not the focus of this article. As is mentioned on page 2, lines 37-39 ("The aim of this article is to present the state of the art for spatio-temporal analyses of droughts derived from documentary evidence for a special issue of the Climate of the Past entitled "Droughts over the centuries: What can documentary evidence tell us about drought variability, severity and human responses."), our overview paper concentrates only on droughts which are identified based either completely or at least partly on documentary data (see the list in Section 2). We checked all of the papers you listed and did not find any that are based/working on/with documentary data in the sense of the definitions in Section 2. Once again, we express our great appreciation to your effort related to collection of all above references. With many thanks, the team of authors

---

## Author Comment (AC2) · 30 Oct 2018

This is a comprehensive review that covers an important area within historical climatology. Whilst it does not contain any new data it goes over a lot of ground and I think will be a useful benchmark for the state of the discipline, hopefully encouraging more research in this area. The synthesis of information from different parts of the world is particularly helpful. RE: We would like to thank the anonymous referee #1 for the generally positive evaluation of our manuscript and the number of constructive comments/suggestions, which we address below.

My comments are mostly relatively minor, although there are a number of them as this review goes over a lot of ground. The only relatively major change I would like to see is much more attention paid to the relationship between spatial variability on droughts and forcings, as this could say something quite interesting about long-term changes in climatologies. As it stands section 5.1 is pretty short and is not integrated with section 4.3. You can probably lose some detail elsewhere if the word count is an issue. RE: Accepted, we are trying to follow the suggestions of the referee through changing various aspects in Sect. 5.1 as specified in our responses to your comments mentioned below.

Abstract: Are historical-climatological databases really a source? RE: Accepted and corrected. We deleted historical climatological databases from the abstract as a documentary source. In Sect. 2 we deleted a separate title 2.15. A corresponding new paragraph dealing with databases is introduced as follows: "Several historical climatology research groups have developed electronic databases containing information about drought and related phenomena (Decker, 2018). Most were initiated due to the need to manage and analyse large amounts of historical-climatological data, and to make these data accessible to international researchers. The first such database, . . ."

p. 1 line 30 - Link between drought and desertification could be made a little clearer. I'm not sure it is enough just to say that they are 'related' – a period of droughts does not necessarily imply desertification. RE: Accepted. The sentence was changed as follows: "One of the related environmental phenomena associated with more frequent and severe drought can be desertification, which not only has an impact on the environment but may also have severe consequences for human society (e.g. Trnka et al., 2018)."

p. 1 line 33 - Give more detail on the 4 types of droughts. RE: Accepted. The second paragraph of Sect. 1 Introduction was changed as follows: "Heim (2002) divided

[Figure]

droughts into four categories: a) meteorological, b) agricultural, c) hydrological and d) socio-economic. Meteorological drought is caused by a direct significant reduction of precipitation totals at the scale of weeks or months compared to mean precipitation patterns in a given area. It can be enhanced by other meteorological factors like air temperature, air humidity or wind speed. Meteorological drought precedes other drought types. Agricultural drought is typically associated with a lack of water for plant growing, and may last from several weeks to 6–9 months; forest stands may also be influenced by drought at the same time scales. Hydrological drought is characterised by a shortage of water in water courses, reservoirs or aquifers, typically lagging behind meteorological and agricultural drought by weeks or months. Socio-economic drought occurs when the negative effects of drought appear in the whole society, influencing everyday life, and socio-economic activities. Fig. 1 shows the sequence of drought occurrence and impacts for generally-accepted types. Mishra and Singh (2010) add underground water drought to these types, as a separate category of hydrological drought. Recently, Van Loon et al. (2016a, 2016b) ..."

P. 2 line 29 – why only reference the Africa consortium? RE: This reference was used in the following context: "For a still more recent world-wide overview of hydroclimate reconstructions based on tree-rings and other natural proxies, the reader is referred to PAGES Hydro2k Consortium (2017); for African hydroclimatic variability over the past 2000 years see Nash et al. (2016a)." It is, we are quoting here overview papers of the worldwide (PAGES Hydro2K Consortium, 2017) or continental scale as for Africa (Nash et al., 2016a). We are not aware of additional publications.

Section 2.1 – you can probably lose some of the quotes here which get a little repetitive. A little more detail on the types on information included would be better. RE: The quotes are the best tools for clearly indicating what type of and how precise evidence can be provided with respect to drought information. Therefore, those texts which may seem repetitive to the general reader provide additional and complementary information in the different sections. Therefore it is necessary that overlaps exist between the

type of information content. The multiple quotations regarding 1-1 source types are purposefully presented to show the variety or similarity of texts and information content in the various continents and very different societal/temporal contexts. Thus, we believe that all these quotations are essential for the understanding of sources and should be kept in each case. In addition, following the suggestion of the reviewer, we are adding some information as in Sect. 2.1: " Annals, chronicles, memoirs and inscriptions are narrative sources that describe with differing degrees of detail mainly important weather/climatic anomalies (including droughts), most commonly those that were outstanding from the point of view of human memory or their impacts on human society. They may contain specific information about periods without rain, dry weather or drought, reporting also lack of water for different human activities, impacts on the everyday life of people, and various other socio-economic impacts or human responses to droughts."

Section 2.3 – also too much focus on quotations and not enough information on what economic records would tell us that others wouldn't. I imagine these would be quite unique, so some specific detail would be helpful. RE: Accepted and corrected by adding following sentences in the first paragraph as follows: "Financial and economic-administrative sources consist of documents prepared at various levels of governmental or state administration. Compared to narrative sources, they contain much more practical, exact or numerical information (in addition to descriptive evidence) on socio-economic impacts, such as the low quantity (or worse quality) of harvests, and the extent of other negative consequences such as the level of food shortage, loss of domestic animals, etc. Further, the different economic aspects of drought may be listed in a more systematic way than in other types of sources, giving an opportunity to obtain additional information about less outstanding drought events. For example, damage to agricultural production ..."

Section 2.7 – more detail is needed here. When were these songs collected? How do we know whether the dates are correct? Given that they are in the folk tradition,

how do we know that the text has not changed over time? RE: It seems, that title "Marketplace and shopkeepers' songs" does not exprees the information values of two songs quoted. From this reason we changed it to "Songs", which indicates, that they are less accurate compared to the other types of documentary evidence. Moreover, we are combining more than one source by cross-checking their content. The first quoted song was published immediately as a response to the drought of 1678, i.e. close to the event, thus it can be considered as a primary source. The second song related to drought in 1790 has the character of a primary source because the author experienced this event personally. It means that the timing of both examples were not a topic of "the folk tradition" with a possible distortion of information being preserved only in folk memory. We hope that the corresponding paragraph contains all basic information to understand this type of sources and examples.

Section 2.8 – again, it would be better to discuss the types of descriptions that are found in newspapers than just to give a long quotation. Newspapers are of course exceptionally diverse, but are there any commonalities in descriptions of drought impacts? RE: Accepted and corrected by including additional information in the first part of the corresponding paragraph as follows: "Droughts and their impacts are types of hydrometeorological extreme frequently reported in newspapers and magazines. Typical accounts in newpapers relate to the economic impacts of drought (especially with regard to cereal harvest expectations and results, implications for animal husbandry, food shortage and other related damage and losses), and the hydrological impact of droughts on water bodies. This information is mostly presented in descriptive or visual (sometimes quantitative) form. However, a range of other types of information may also appear in newspapers and journals. The validity and importance of such sources . . ."

Section 2.9 – could you provide a little more information on the way that these 'winter counts' were recorded? RE: Accepted and corrected. The corresponding paragraph was changed as follows: "Gallo and Wood (2015) used Native American pictographic evidence (known as "winter counts") for the study of past drought events in the U.S.

Great Plains. Starting with annual documentation of significant events from the late 1600s, these counts evolved into drawings or pictographic records of the major events of the year (see Fig. 1 in the quoted paper). Winter counts were related to the period between the first snowfalls of successive years. Winter counts were the responsibility of a specific keeper who would consult with a council of elders as to which events were important enough to be recorded. From these counts, included in five printed or on-line documents, Gallo and Wood (2015) identified ten prolonged dry or drought events between 1700 and 1880 that correspond with other observations or available information (particularly with PDSI values from North American Drought Atlas by Cook and Krusic, http://iridl.ldeo.columbia.edu/SOURCES/.LDEO/.TRL/.NADA2004/.pdsi-atlas.html, last access: 27 August 2018)."

Section 2.10 – a little more detail would be helpful here. Where and when were these records collected? Why were they expressed in this way? RE: Accepted and corrected in the first sentence of the corresponding paragraph as follows: "Chronograms are paintings or carvings depicting memorable events (years) that significantly affected society (including droughts) for common/public remembrance/memory. Many were painted on walls of buildings or carved on stone statues, but could also form a part of chronicles or other narratives, at least from the Antique period onwards."

Section 3 (iii) – any examples from outside of the Czech Lands? RE: We are not aware of any reconstruction of standard drought indices (SPI, SPEI, Z-index or PDSI) based on documentary data outside the Czech Lands.

Page 20 line 36 – personal bugbear here. It is fine to say that the 'Southern Oscillation (SO) made only minor contributions to central European drought variability', but not the 'Southern Oscillation Index'. The index is just a set of numbers that represents the climatic forcing: the Southern Oscillation or ENSO. Suggest you replace SOI with SO throughout the text. RE: Accepted and corrected. SOI was changed to SO in the whole manuscript.

I'm a little unsure about 'positive values of the NAO had a strong effect on a December–August drought index' as well, but at least here you are associating an index with another index, rather than an index with a drought. RE: We changed 'NAO' to 'NAOI' in this sentence. We are here simply quoting the main finding of the paper by Vicente-Serrano et al. (2007).

Page 24 line 12-25 – I'm not sure about this paragraph and I think you should remove it entirely. There's nothing particularly significant about using documents to look at droughts during the 'instrumental period'; it may be unusual within historical climatology but there's a huge literature on 20th century droughts from outside of historical climatology, and the instrumental record itself of course tells us very little about drought impacts. You'd either have to cite a far larger literature than you do here (Sen? Mukerjee?McAlpin? – and that is only India) or stick to what historical climatology can contribute, i.e. pre-instrumental. RE: Many documentary data continue into the instrumental period and may provide significant additional information that is not collected as part of meteorological measurements. Despite the papers quoted in this part of the manuscript concerning the instrumental period, they were based on documentary data. For this reason we included and discussed them; we are not including comprehensive lists of papers dealing with droughts based on meteorological/hydrological measurements, as this would be beyond the scope of this article.

Section 4.4 – I wonder if there is a way to restructure this section so that you don't end up with a subsection on 'ancient civilisations and colonial regimes'. I don't quite see the association between the Neo-Assyrian Empire, colonial Mexico and colonial Australia. I suggest removing this section altogether. The first and third paragraphs and the material on the Pueblo could go into section 4.4.1 and the material on Mexico into section 4.4.4. RE: Accepted. We thank referee #1 for a very useful suggestion: we removed Section 4.4.2 and changed the sections accordingly.

Section 5.1 – I would like to see more of a discussion of the spatial extent of droughts in relation to the climatic forcings discussed in section 4.3. This is where documentary

evidence could be particularly useful, for example to discuss how teleconnection patterns have changed over time. I know that Nash has done this a few times for southern Africa, for example. RE: The paragraph concerning Europe is presented below as a response to your comment on page 28. Changes made in the paragraph related to China are as follows: "Turning to the long-term decadal fluctuations of documentary-based drought characteristics in China, Fig. 10 shows drought variability over the two past millennia for the series spanning area of eastern China with a prominent monsoonal climate: (i) decadal means of regional dry-wet index series for the North China plain (c. 34–40°N, east of 105°E), the Jiang-Huai area (c. 31–34°N, east of 110°E), the Jiang-Nan area (c. 25–31°N, east of 110°E) and (central) Eastern China (c. 25–40°N, east of 105°E), derived from an annual drought/flood grade dataset after detrending the effects of absent data on homogeneity (Ge et al., 2014, 2016); (ii) decadal means of degrees of dryness/wetness in the Great Bend of the Yellow River region (Yang et al., 2014); (iii) decadal frequencies of droughts in the Yangtze Delta (Jiang et al., 2005). It should be noted that the North China, Jiang-Huai and Jiang-Nan series (Fig. 10a–c) are all geographically situated within the area encompassed by the Eastern China series (Fig. 10d). As such, it is not surprising that the Eastern China series exhibits the highest correlation coefficients with the dry-wet index series of its three parts (0.81 with North China, 0.79 with with the Jiang-Huai area and 0.62 with the Jiang-Nan area), and marginally lower correlations with the two remaining series (0.57 with the Yellow River region and −0.42 with the Yangtze Delta). The dry-wet index series correlate better between neighbouring areas (r = 0.52 for North China versus the Jian-Huai area; 0.46 for Jian-Huai area versus the Jiang-Nan area), while for the two more distant regions the figure is only 0.18. The series for the Great Bend of the Yellow River region is best correlated with North China (r = 0.67) and the series of drought frequencies in the Yangtze Delta with the Jiang-Huai area (r = −0.46) due to the territorial overlap of regions analysed. All these correlation coefficients are statistically significant at the 0.05 significance level (a non-significant correlation coefficient of 0.11 exists only between the Jiang-Nan area and the Yellow River region). Correlation coefficients in the eastern

China series can be influenced by the spatial patterns of droughts/floods, as shown by Hao et al. (2016) from the analysis of cold and warm periods at a centennial scale. Cold periods show an east-to-west distribution, with floods east and droughts west of 115°E. In the case of warm periods a tri-pole patterns appears, with droughts south of 25°N and north of 30°N, and floods in between (i.e. from 25–30°N). As noted in Sect. 4.3.2, weakening of the Asian summer monsoon and anomalous displacement of the western Pacific subtropical high can be important triggers of drought episodes in eastern China (Shen et al., 2007; Yang et al., 2014). Droughts can be also correlated with the phases of ENSO (Ji-bin et al., 2006). However, the situation is not simple, as both La Niña and El Niño phases may be associated with droughts, as shown for different parts of the Yangtze River (e.g. Jiang et al., 2006; Tong et al., 2006). Several papers also mention relationships between past Chinese droughts and volcanic eruptions (e.g. Shen et al., 2007; Ge et al., 2016) or even to solar activity (e.g. Shen et al., 2008; Wan et al., 2018). New reference: Hao, Z., Zheng, J., Zhang, X., Liu, H., Li, M., and Ge, Q.: Spatial patterns of precipitation anomalies in eastern China during centennial cold and warm periods of the past 2000 years, Int. J. Climatol., 36, 467–475, doi: 10.1002/joc.4367, 2016.

The following paragraph is added after the paragraph related to Africa: "A number of studies have explored the factors influencing the timing of historical African droughts (see also Sect. 4.3.3). Much of this work has centred upon southern Africa, a region with strong ENSO teleconnections. Lindesay and Vogel (1990) were one of the first authors to identify the historical association between El Niño and drought in the Cape coastal zone of South Africa. This association has now also been noted in the Kalahari (Nash and Endfield, 2002a), Lesotho (Nash and Grab, 2010), KwaZulu-Natal (Nash et al., 2016), Malawi (Nash et al., 2018) and central Namibia (Grab and Zumthurm, 2018). It must be stressed that not all droughts in the subcontinent are associated with El Niño events. Further, not all El Niño events are followed by droughts. When El Niño-related droughts do occur, their spatial impact depends upon the geographical distribution of the ENSO influence. This is best illustrated with reference to Malawi, which is located

between two regions of opposing climatic response to El Niño – eastern Equatorial Africa to the north often receives above average rainfall following an El Niño event, while areas to the south experience drought. Following the moderate to very strong 1877–1878 El Niño event, severe drought occurred across the entire southern African subcontinent, including Malawi (Fig. 11). In contrast, during the moderate to strong 1896–1897 event, Malawi experienced very wet conditions while the remainder of the subcontinent suffered from severe drought (Nash et al., 2018)."

The paragraph related to the Americas was extended as follows: "Drought series available for the Americas relate largely to the past 500 years (Fig. 12). The following drought series are available: (i) decadal frequencies of droughts in central Mexico (Mendoza et al., 2005a); (ii) decadal frequencies of very dry and dry years on the Pacific coast of Central America (Guevara-Murua et al., 2018); (iii) decadal frequencies of dry and very dry years for Potosí, Bolivian Altiplano, Bolivia (Gioda and Prieto, 1999). The differences in the fluctuation of drought frequency in all three areas are reflected in their low and statistically non-significant correlation coefficients: between (i) and (ii) 0.27, (i) and (iii) 0.07, (ii) and (iii) –0.22. This reflects the totally different effects of large-scale climate drivers influencing drought occurrence in three selected, very remote areas."

Page 28 lines 32-34 – this analysis seems incomplete. What about the relationship between the Czech series and the Soviet Union, or Spain and western Europe? Why might some of these series be related but others not? RE: This analysis was complemented in the corresponding paragraph as follows: "Fig. 9 shows a selection of long-term decadal fluctuations in several documentary-based drought characteristics related to Europe over the past millennium, extracted from a range of papers: (i) frequency of dry months in Western Europe from chronicle sources (Alexandre, 1987); (ii) standardised weighted index for drought rogation ceremonies for four sites (Barcelona, Girona, Tarragona, Tortosa) on the Catalonian coast of north-eastern Iberia (Oliva et al., 2018); (iii) frequency of droughts and PDSI over the territory of the Czech Lands

(Brázdil et al., 2013, 2016a); (iv) 30-year frequencies of extremely dry MAM–JJA seasons in the European non-chernozem part of the former Soviet Union (Lyakhov, 1984). While both the Czech series are highly correlated (r = –0.65, statistically significant at the 0.05 significance level), no relationship can be found between the Czech drought series and the drought rogation index of the Catalonian coast of north-eastern Iberia (insignificant correlations –0.13 and –0.12 respectively). This clearly follows from the climatic differences existing between the Mediterranean and central European region, located in different climatic zones under the influence of different large-scale climate drivers. For example, despite the fact that the effects of the NAO are important both in the Czech Lands (Mikšovská et al., 2018) and in Spain (Vicente-Serrano and Cuadrat, 2007), changes in circulation patterns between these two regions for positive and negative phases of NAO are fundamental (e.g. Hurrell, 1995; Wanner et al., 2001). The comparison of droughts in the European non-chernozem part of the former Soviet Union (Lyakhov, 1984) with Western European, Spanish and Czech drought series, transformed to 30-year periods, shows that statistically significant correlations were found only with Spain drought rogation ceremonies (r = –0.59) and Czech PDSI (r = 0.64). Positive, although non-significant, correlation was found also with a frequency series of dry months in Western Europe (r = 0.41). This could perhaps signal some consistency of drought occurrences in the same latitudinal belt under the prevailing influence of western airflow; however, the smaller temporal resolution of drought data due to the considerable smoothing of data may have produced artificially higher correlations." New references: Hurrell, J. W.: Decadal trends in North Atlantic Oscillation regional temperatures and precipitation, Science, 269, 676–679, doi: 10.1126/science.269.5224.676, 1995. Wanner, H., Brönnimann, S., Casty, C., Gyalistras, D., Luterbacher, J., Schmutz, C., Stephenson, D. B., and Xoplaki, E.: North Atlantic Oscillation – concepts and studies, Surv. Geophys., 22, 321–382, doi: 10.1023/A:1014217317898, 2001.

Section 5.2 – The Cook et al. megadrought reconstructions should be mentioned here in comparison with documentary data. That team have done at least three now (North

America, Monsoon Asia and Europe). Given that they're calculated only using tree rings and statistical analysis, documentary data could provide a useful verification or comparison. RE: To perform a comparison with Cook et al. gridded drought reconstructions (scPDSI) for North America, Monsoon Asia and Europe with documentary-based droughts would need a separate study where one would first calculate similar tree-ring based PDSI series for regions in which any documentary-based drought series are available (as was done, for example, for the Czech Lands or central Europe in Brázdil et al., 2016a, Fig. 8). This is rather difficult and clearly beyond the scope of this review article, where we simply mention papers in which any comparison with Cook et al. reconstructions have been already done (see, e.g., for Europe on page 20, lines 3–7 and page 30, lines 28–29; for Monsoon Asia-China see page 30, lines 30–34; for North America it was added to Sect. 2.9 – see our response above, where Gallo and Wood, 2015, used such data for interpretation of "winter counts"). Similar comparison, but again for the Czech Lands and with Z-index, is prepared in the paper by Dobrovolná et al. (2018), which is more-or-less ready for submission to SI CoP. Because it was not yet submitted, we do not quote any results of this study in our article, showing only a corresponding Fig. 1:

Fig. 1: (a) Long-term variability of reconstructed AMJJ Z-index (based on combined documentary and tree-ring data) for the Czech Republic compared to scPDSI (Cook et al., 2015, areal mean of the 48–51°N; 12–19°E region) in the 1501–2012 period; original series were smoothed by 30-year Gaussian filter; (b) 30-year running correlations between the Z-index and PDSI series; correlations above the dashed horizontal line are statistically significant
* * *
[Figure]

**Fig. 1.**

---

## Author Comment (AC3) · 30 Oct 2018

This extensive manuscript provides a thorough overview of historical drought studies based on documentary evidence for most of the world. The focus is more on Europe and Africa than other areas, but that makes sense given the author list. The article is very long but well written, and I recommend that it be published once the authors have considered the following points:

RE: We would like to thank the anonymous referee #2 for very useful critical comments

and suggestions. Apart from the authors' list, the reason for having more papers related to Europe is simply that the greatest volume of documentary-based research into historical droughts is produced for Europe. Regarding Africa, intense research is found in southern Africa, while the rest of the continent is generally underrepresented. Thus, the proportions in the paper represent the different density of scientific research in the various areas of the world.

Major comments: - There's no real discussion in this article about the limitations of documentary sources when it comes to climatological analysis - the impact of non-climatic elements in many of the evidence types mentioned in section 2, and the subjective nature of human memory. Will this be covered in a separate article in the special issue? If not, I think it is worth adding a paragraph or two about it in the discussion. If there will be another article focussed on this, please mention it. RE: No, nor do we plan an article on this topic. We believe that we have clearly pointed out in the second paragraph of Sect. 3 the key points of historical-climatological analysis: (i) use of primary sources, contributing to avoiding erroneous data, (ii) source-critical approach to eliminate effects of non-climatic factors, (iii) cross-checking of data from a spatial and temporal point of view, (iv) careful analysis and interpretation of evidence available. We know that "the subjective nature of human memory" is sometimes used as an argument against the value of documentary data, but it has no importance in working with documentary evidence if the above mentioned points are applied. The present article aims to provide an overview of existing papers and knowledge, and does not really intend to discuss methodological problems of historical climatology or cover and solve all current questions and problems of a research area (see e.g. extensive references to this topic – at least Brázdil et al., 2005, 2010). We have reworded the corresponding paragraph to address your comment: "The extraction of drought information from documentary sources requires a source-critical approach generally applied to scientific work in historical climatology (for more detailed discussion see e.g. Brázdil et al., 2005, 2010). It includes the following important steps: (i) the use of primary sources to avoid possible errors that may appear in secondary sources (e.g. weather compilations as mentioned

in Sect. 2.14, or information from heresay, i.e. events not directly experienced by the author); (ii) a source-critical approach to eliminate the effects of non-climatic factors (e.g. to avoid possible 'social bias', taking in account broader socio-economic knowledge related to the given source); (iii) cross-checking of data from spatial and temporal perspectives by combining different types of documentary sources; (iv) careful meteorological (climatological) interpretation and analysis of the available evidence based on knowledge of recent climatic patterns in the area. Applying these principles allows the true spatial extent, duration, severity and impacts of individual drought events to be identified."

- It's not clear to me why the Americas and Australia are grouped together in section 4.1 and 4.3. Drought responses there are almost anti correlated, particularly rainfall variability driven by ENSO. Is it for colonial reasons, or because similar methods are used? If so, please state that explicitly. You also mention New Zealand in section 4.4.3 (page 26), but this is not included elsewhere. It would be better to split America and Australia into two separate parts: The Americas, and the South Pacific, including Australia, NZ and possibly Pacific Island studies (if there are any). RE: The reason for such division is based on the simple fact that the corresponding papers dealing with documentary-based droughts are much less frequent compared to Europe, Asia (China) or Africa. By splitting "The Americas and Australia" into two parts, Australia (or as the reviewer suggests South Pacific) would be represented by only one paragraph in Sect. 4.1 (page 18) or only one sentence in Sect. 4.3 (page 22). We are not aware of any published studies on historical drought from New Zealand or the Pacific Islands. For this reasons it seems to be more reasonable to have The Americas and Australia together as "a remaining regions of the world".

-Somewhere in section 2, I wonder if you could mention information that falls between documentary and instrumental. Things like counts of rain days derived from weather journals and newspapers, or crop yield totals. I don't think this needs its own section, but could be slotted into others to show that it's not only words that can be useful.

RE: In Section 2 we reported the basic types of documentary evidence which cover the pre-instrumental period. Many of these data sources continue in the instrumental period. It seems that there is not "anything" between documentary and instrumental data/periods. For example, counts of rainy days derived from some types of documentary sources described in Sect. 2 can be used for the creation of precipitation indices. We see the use of crop yield totals or grain prices as more problematic since, without additional information, such data do not necessarily express the real effects of droughts.

-Section 4.2 confused me a bit. Are these the biggest events to be found in the many paper listed in section 4.1, or large overarching droughts that affected many countries? I think it's the latter, but this could be clarified with an introductory sentence or two, or by reshaping the section to focus on the timing of events. You could even tighten this section, removing reference to droughts that only occur in one country. RE: To explain our motivation for separating both sections: In Sect. 4.1 "Long-term precipitation and drought series" we present studies dealing with long-term series of droughts. The following Sect. 4.2 "Individual and major droughts events" aims at presenting contributions that discuss individual (important) drought episodes or only drought cases that do not represent long-term chronologies as in Sect. 4.1. To distinguish between both sections, we added the following sentence at the beginning of Sect 4.2: "While the previous section (Sect. 4.1) concentrated on papers dealing with long-term fluctuations in droughts, this section reviews studies oriented towards complex analyses of either one particular extreme drought event with its human consequences or a few such severe drought episodes. For example, Pankhurst (1966) reported 1888, a year of major El Niño, as excessively dry and hot in Ethiopia, ..."

Minor things: -Page 2, line 38: I don't think you need 'the' before Climate of the Past RE: Accepted and corrected.

-Page 4, line 32: 'Related legal trials' instead of 'Legal trials related' RE: Accepted and corrected.

-Page 4, line 33: 'fashions' rather than 'fashion' I think RE: Accepted and corrected.

-Page 9, line 27: using numbers, hyphens and minus signs together is confusing, can you use an equal sign or colon instead? RE: Accepted and changed as follows: "For instance, 3-degree (–1: dry, 0: normal, 1: wet), 5-degree (–2: very dry/drought, –1: relatively dry, 0: normal, 1: relatively wet, 2: extremely wet) or 7-degree (–3: extremely dry, –2: very dry, –1: dry, 0: normal, 1: wet, 2: very wet, 3: extremely wet) scales are the most widely used in Europe (e.g. Pfister, 1992, 1999, 2001; Glaser, 2001, 2008; Dobrovolná et al., 2015a) and Africa (e.g. Nicholson et al., 2012a, 2012b; Nash et al., 2016b, 2018)."

-Page 13, line 9: As above RE: Accepted and changed as follows: "A 5-degree scale was used for classification: 1: very wet, 2: wet, 3: normal, 4: dry, and 5: very dry."

-Page 13, line 27: 'Droughts were more extreme in these centuries than in the 20th century'. This sentence could be clarified. RE: Accepted. The corresponding part of the manuscript was changed as follows: "Wang et al. (2015), using documentary-based drought data from Eastern China for the period 1470–2000, reported a higher number of droughts during the 16th and 17th centuries than in the 18th and 19th centuries. Droughts were more extreme in these four centuries than in the 20th century."

-Section 4.1.2: No mention of Japan? RE: We did not find any Japanese paper dealing with droughts based on documentary data. Moreover, communication with a leading Japanese historical climatologist, Prof. Takehiko Mikami (former Tokyo Metropolitan University) revealed, "no systematic papers on drought history in Japan have been published" and "drought disasters in historical times were much less than flood disasters in Japan."

-Page 15, line 6: The term Nilometer could be explained better RE: Accepted and corrected. The corresponding sentence was complemented as follows: "While, strictly speaking, an indicator of rainfall over the Nile catchment areas in Ethiopia (Blue Nile) and equatorial Africa (White Nile), Nilometer records from Cairo (stone structures at

which levels of the River Nile were recorded with respect to a vertical column, a series of steps leading down to the river, or a deep well with culvert; see Popper, 1951) also provide a near-annually resolved drought chronology for north-eastern Africa dating back to the 7th century."

-Page 18, line 21: 'most severe' rather than 'severest' I think RE: Accepted and corrected.

-Page 20, line 32: 'above effects' is unclear to me RE: Accepted. The corresponding sentence was changed as follows: "Relatively few studies have investigated the effects of external forcing and large-scale climate drivers for drought series in Europe (e.g. Pongrácz et al., 2003)."

-Page 22, line 27: 'Ashcroft' rather than 'Aschcroft' RE: Accepted and corrected.

-Page 23, line 14: thank you for teaching me the word 'transhumant', I've never seen it before RE: OK.

-Page 24, line 12: add 'the' before 'instrumental period' RE: Accepted and corrected.

-Section 4.4.1, final paragraph: I like this qualifier, and am sure there are many other sources of information about the pros and cons of environmental determinism. Perhaps the authors could provide another overarching reference to this topic? RE: The paragraphs and the chapter are not built to discuss environmental criticism to any more specific level, and we do not intend to go into the problems of the rather broad and well-discussed field of environmental determinism in more detail. To start referring to papers on general environmental determinism would suggest that we also intend to deal with this problem in more details in this paragraph or in the paper itself.

-Page 25, line 13: errant bracket RE: Accepted and corrected.

-Page 25, line 36: 'most immediate impact' on what? RE: Accepted and corrected as follows: "For this reason, such events may have the greatest and most immediate impact on society (e.g. in water supply and food production) of all climate changes."

[Figure]

-Page 28, line 21-22: this sentence is a bit clunky and could be rearranged RE: Accepted. The corresponding sentence was changed as follows: "But corresponding papers differ in the density and quality of documentary data used for identification of droughts, in definitions and selection of droughts, in the areas analysed, as well as in the time periods covered."

-Page 28, line 24: add 'well' before correlated RE: Accepted and corrected.

-Page 31, line 23: remove the 's' from sources RE: Accepted and corrected.

-Section 6: it sounds like another area for concerted effort is cross regional comparisons of historical droughts. RE: Accepted and corrected. We added it to the point (ii) as follows: "Cross-regional comparisons of past droughts."
* * *
[Figure]

---

## Author Comment (AC4) · 30 Oct 2018

The paper "Documentary data and the study of the past droughts: an overview of the state of the art worldwide" by R. Brazdil et al. aims at presenting the state of the art for spatial-temporal analyses of droughts derived from documentary evidence. It gives an excellent overview of the topic discussing types of documentary sources, methods for reconstructing droughts from them, long-term drought series and related forcings and impacts and in my opinion it will give a valuable contribution to the special issue

in which it is going to be published. The overall quality of the paper is excellent: even though it is very long it does not contain unnecessary information as the length of the paper reflects the huge amount of work the authors performed to review the state of the art of past drought analyses worldwide. This huge amount of work is also clear from the long and complete list of references that is provided in the paper. This paper will really be a milestone for anyone interested in past drought reconstruction. The paper is also well-organized which makes it easy to read in spite of the very large amount of information that is provided. RE: We would like to thank the anonymous referee #3 for the generally very positive evaluation of our manuscript.

I therefore think that the paper can be published in its present form and the only minor suggestion I give to the authors is to consider whether it is really necessary separating section 4.2 from section 4.1. The differences between the events discussed in these two sections are in fact not completely clear and the events discussed in sections 4.2 could also be moved to section 4.1. RE: To explain our motivation for separating both sections: In Sect. 4.1 "Long-term precipitation and drought series" we present studies dealing with long-term series of droughts. The following Sect. 4.2 "Individual and major droughts events" aim at presenting contributions that discuss individual (important) drought episodes or only drought cases that do not represent long-term chronologies as in Sect. 4.1. To distinguish between both sections, we added the following sentence at the beginning of Sect 4.2: "While the previous section (Sect. 4.1) concentrated on papers dealing with long-term fluctuations in droughts, this section reviews studies oriented towards complex analyses of either one particular extreme drought event with its human consequences or a few such severe drought episodes. For example, Pankhurst (1966) reported 1888, a year of major El Niño, as excessively dry and hot in Ethiopia, ..."